# The Devil is in the Condition Numbers:
# Why is GLU Better than non-GLU Structure?

**Xingyu Lyu** [1 2]   **Qianqian Xu** [1 3]   **Zhiyong Yang** [2]   **Peisong Wen** [2]   **Qingming Huang** [2 1]

## Abstract

Gated Linear Units (GLU) and their variants are widely adopted in modern open-source large language model architectures and consistently outperform their non-gated counterparts, yet the underlying reasons for this advantage remain unclear. In this work, we study GLU by analyzing two-layer networks in the neural tangent kernel (NTK) regime. Our analysis reveals that the GLU structure reshapes the NTK spectrum, leading to a smaller condition number and a more compact eigenvalue distribution. Building on this finding, we further analyze the resulting training dynamics and show how the reshaped spectrum leads to faster convergence of GLU models, including a characteristic loss-crossing phenomenon observed between GLU and non-GLU models. Finally, we empirically observe that GLU has limited impact in reducing the generalization gap on various models, including ViT and GPT-2, suggesting that its primary benefit lies in accelerating optimization rather than reducing the generalization gap. The code is available at: https://github.com/Zemdalk/GLU-NTK.

## 1. Introduction

Gating mechanisms have a long history in neural network architectures as an effective means to improve optimization and expressivity (Hochreiter & Schmidhuber, 1997; Dey & Salem, 2017). In modern open-source large language models, Gated Linear Units (GLU) and their variants, most notably SwiGLU, are widely adopted in feed-forward

---

[1]State Key Laboratory of AI Safety, Institute of Computing Technology, Chinese Academy of Sciences, Beijing 100190, China [2]School of Computer Science and Technology, University of Chinese Academy of Sciences, Beijing 101408, China [3]Beijing Academy of Artificial Intelligence (BAAI), Beijing, China. Correspondence to: Qianqian Xu <xuqianqian@ict.ac.cn>, Qingming Huang <qmhuang@ucas.ac.cn>.

*Proceedings of the 43rd International Conference on Machine Learning*, Seoul, South Korea. PMLR 306, 2026. Copyright 2026 by the author(s).

network (FFN) blocks due to their superior empirical performance (Shazeer, 2020; Yang et al., 2025; Wang et al., 2025). The GLU structure is characterized by a remarkably straightforward form:

$$\text{GLU}_\phi(\mathbf{x}) = (\mathbf{Px}) \odot \phi(\mathbf{Wx}),$$

where $\mathbf{W}, \mathbf{P} \in \mathbb{R}^{m \times d}$ are learnable weight matrices, $\phi$ is the activation function and $\odot$ denotes the Hadamard multiplication.

Empirically, even in simple two-layer network settings, introducing GLU leads to consistent performance improvements over non-gated counterparts (see Fig.6). Similar gains have also been observed when GLU-style gating is applied to attention mechanism (Wang, 2025). These observations indicate that the advantages of GLU are not confined to a particular architectural setting, suggesting a fundamental benefit inherent to the gating structure itself. However, the theoretical reasons behind the advantages of GLU variants remain unclear.

We aim to systematically investigate the origin of these gains. Specifically, we must distinguish between two factors. First, GLU might offer more efficient fitting on training data. Second, it might be able to better reduce the gap between training and evaluation loss. To this end, we formalize our study by decomposing the population loss $\mathcal{L}_\mathcal{D}(f_\boldsymbol{\theta})$ into *training error* and *generalization gap*:

$$\underbrace{\mathcal{L}_\mathcal{D}(f_\boldsymbol{\theta})}_{\text{generalization error}} = \underbrace{\mathcal{L}_S(f_\boldsymbol{\theta})}_{\text{training error}} + \underbrace{(\mathcal{L}_\mathcal{D}(f_\boldsymbol{\theta}) - \mathcal{L}_S(f_\boldsymbol{\theta}))}_{\text{generalization gap}}.$$

Here $\mathcal{D}$ represents the underlying data distribution, $S$ is the training set sampled from $\mathcal{D}$, and $f_\boldsymbol{\theta}$ is the model parameterized by $\boldsymbol{\theta}$. Intuitively, the training error term captures how well the model fits the training data, whereas the generalization gap term describes how well the model learns the inherent structure from the training data. This decomposition naturally leads to two questions: *(1) how do GLU variants affect optimization behavior during training?* And *(2) do they influence generalization gap?*.

In short, we find that **(1): GLU structure accelerates the convergence of training error; (2): the generalization gap stays roughly the same as the non-GLU variants;**

**(3): (1) and (2) lead to a better overall generalization error of GLU.**

**For the training error, we find that GLU accelerates the convergence of training process in the Neural Tangent Kernel (NTK) regime**. While directly analyzing neural network optimization is inherently challenging, the NTK framework provides theoretical tractability by characterizing optimization through the spectral properties of the associated kernel matrix. Specifically, our main theoretical finding is that, in the kernel regime, the NTK matrix $\tilde{\mathbf{K}}$ induced by GLU structure admits an approximate Hadamard-product structure of the form:

$$\tilde{\mathbf{K}} \approx \mathbf{K} \odot (\mathbf{X}\mathbf{X}^\top/d),$$

where $\mathbf{K}$ denotes the NTK of the corresponding non-GLU model, and $\mathbf{X} \in \mathbb{R}^{n \times d}$ is the input data matrix. As is shown in Fig.1, this leads to a better-conditioned kernel spectrum, characterized by a more contracted eigenvalue distribution (Sec.3). According to existing results in NTK, this clearly suggests that GLU leads to faster convergent kernel regime (de Ryck et al., 2024; Terjék & González-Sánchez, 2025). More interestingly, our result also shares a close connection with the recent work on model gradient angle theory. Specifically, we show that such gating mechanism can be interpreted as enlarging the model gradient angle between $\nabla_{\boldsymbol{\theta}} z(\mathbf{x})$ and $\nabla_{\boldsymbol{\theta}} z(\mathbf{x}')$, making samples more separated in gradient feature space. According to Liu et al. (2025), this yields improved optimization rates for GLU-based models.

Building on this spectral perspective, we further analyze how the modified NTK spectrum affects the training dynamics over time, providing a formal explanation for the stage-wise convergence behaviors and the loss-crossing phenomenon observed in practice (Sec.4).

For the generalization gap, it stays roughly the same as the non-GLU variants. Through empirical comparisons of training loss $L_S$ and generalization gap $L_{\mathcal{D}} - L_S$, we observe that for the same training error, GLU-based and non-GLU models generally have similar generalization gap (Sec.5). This suggests that the primary advantage of GLU lies mainly in accelerating optimization rather than in reducing the generalization gap. Nonetheless, this is sufficient to reduce the overall generalization error.

In summary, our contributions are threefold:

- We provide a theoretical analysis of GLU variants in the kernel regime, showing that GLU structure leads to improved spectral conditioning.

- We analyze the resulting optimization dynamics through the NTK spectrum and further explain characteristic training behaviors of GLU-based models, including stage-wise convergence behavior.

- We empirically examine the effect of GLU on generalization gap and find that, when training with gradient descent algorithm, GLU does not significantly alter the generalization gap, indicating that its primary advantage lies in optimization.

## 2. Preliminary

In this section, we introduce the key concepts used throughout the paper, including Gated Linear Unit (GLU) variants, the neural tangent kernel (NTK), and population loss decomposition.

### 2.1. Gated Linear Units (GLU) and its Variants

Modern large language models (LLMs) widely adopt Gated Linear Unit (GLU), particularly the SwiGLU variant, as the default feedforward block in Transformer architectures (Grattafiori et al., 2024; Liu et al., 2024). For an input $\mathbf{x} \in \mathbb{R}^d$, the GLU structure is defined as (Shazeer, 2020)

$$\text{GLU}_\phi(\mathbf{x}) = (\mathbf{P}\mathbf{x}) \odot \phi(\mathbf{W}\mathbf{x}), \tag{1}$$

where $\mathbf{W}, \mathbf{P} \in \mathbb{R}^{m \times d}$ are learnable weight matrices, $\phi(\cdot)$ is pointwise activation function, and $\odot$ denotes elementwise multiplication.

Different choices of $\phi$ yield different GLU variants, including:

- **ReGLU**: $\phi(x) = \text{ReLU}(x) = \max\{0, x\}$,

- **GEGLU**: $\phi(x) = \text{GELU}(x) = \frac{x}{2}\left[1 + \text{erf}\left(\frac{x}{\sqrt{2}}\right)\right]$,

- **SwiGLU**: $\phi(x) = \text{Swish}_1(x) = x \cdot \sigma(x)$.

Here $\sigma(\cdot)$ represents the sigmoid function, and $\text{erf}(\cdot)$ stands for the error function.

All GLU variants share the same multiplicative gating structure Eq.(1), which will play a central role in our subsequent analysis.

### 2.2. Neural Tangent Kernel (NTK)

To analyze optimization dynamics, we adopt the neural tangent kernel (NTK) framework (Jacot et al., 2018). For a network $f_{\boldsymbol{\theta}}(\mathbf{x})$ parameterized by $\boldsymbol{\theta}$, for any two inputs $\mathbf{x}, \mathbf{x}' \in \mathbb{R}^d$, the NTK function is defined as

$$K(\mathbf{x}, \mathbf{x}') = \langle \nabla_{\boldsymbol{\theta}} f_{\boldsymbol{\theta}}(\mathbf{x}), \nabla_{\boldsymbol{\theta}} f_{\boldsymbol{\theta}}(\mathbf{x}') \rangle.$$

Given training inputs $\{\mathbf{x}_i\}_{i=1}^n$, the empirical NTK matrix $\mathbf{K} \in \mathbb{R}^{n \times n}$ is given by

$$K_{ij} = K(\mathbf{x}_i, \mathbf{x}_j).$$

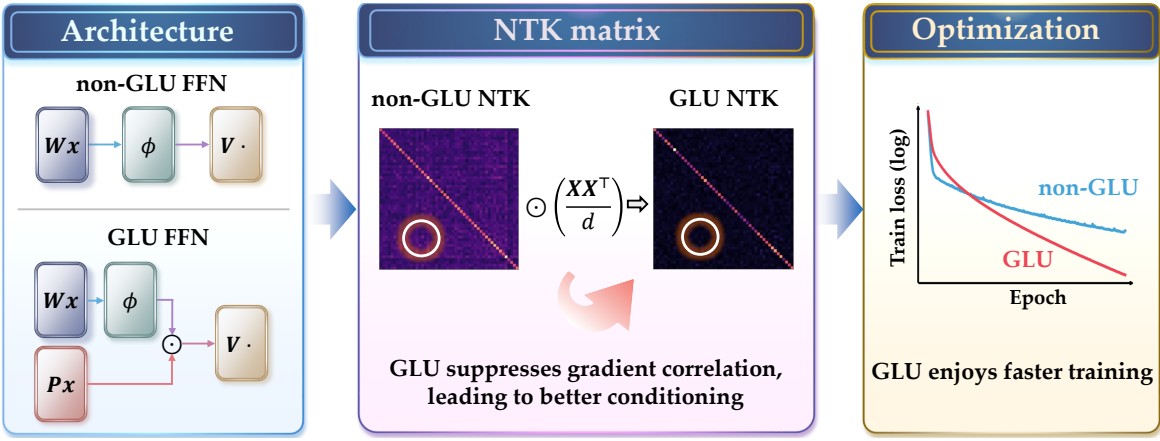

*Figure 1.* **Illustration of our theoretical discovery.** We find that (A) by adding the gating structure, (B) the NTK matrix of GLU structure becomes better conditioned, which (C) explains the faster optimization of GLU-based models.

Let $\lambda_{\max}(\mathbf{K})$ and $\lambda_{\min}(\mathbf{K})$ denote the largest and smallest eigenvalues of $\mathbf{K}$. The NTK condition number is defined as

$$\kappa(\mathbf{K}) = \frac{\lambda_{\max}(\mathbf{K})}{\lambda_{\min}(\mathbf{K})}.$$

A series of recent studies has demonstrated that the conditioning of the NTK, i.e., the condition number, is strongly connected with the convergence rate of gradient descent algorithms(Liu et al., 2020; de Ryck et al., 2024; Liu et al., 2025; Terjék & González-Sánchez, 2025). In particular, Liu et al. (2025) summarize that, the worst-case empirical loss $L_S$ satisfies

$$\mathcal{L}_S(\boldsymbol{\theta}_t) \leq (1 - \kappa^{-1})^t \mathcal{L}_S(\boldsymbol{\theta}_0),$$

where $\boldsymbol{\theta}_t$ is the model parameter set at step $t$, $\kappa$ is the NTK condition number.

Consequently, achieving error $\epsilon$ requires

$$N(\epsilon) = \frac{\log(\epsilon/\mathcal{L}_S(\boldsymbol{\theta}_0))}{\log(1 - \kappa^{-1})} = O\left(\kappa \log(1/\epsilon)\right)$$

iterations, making the NTK condition number the key indicator of optimization speed.

### 2.3. Wishart Matrices and the Marchenko-Pastur Law

In our analysis, we will make use of classical results from random matrix theory concerning Wishart matrices. Let $\mathbf{X} \in \mathbb{R}^{n \times d}$ be a data matrix whose rows are independent samples with zero mean and identity covariance. The corresponding (scaled) sample covariance matrix

$$\mathbf{S} = \frac{1}{d}\mathbf{X}\mathbf{X}^{\top} \qquad (2)$$

is referred to as a Wishart matrix.

When both $n, d \to \infty$ with $n/d \to \gamma \in (0, \infty)$, the empirical spectral distribution of $\mathbf{S}$ converges almost surely to the Marchenko-Pastur distribution (Marčenko & Pastur, 1967). The Marchenko-Pastur distribution has density

$$\rho_{\mathrm{MP}}(\lambda) = \frac{\sqrt{(\lambda_+ - \lambda)(\lambda - \lambda_-)}}{2\pi\gamma\lambda} \, \mathbf{1}_{[\lambda_-, \lambda_+]}(\lambda),$$

where $\lambda_{\pm} = (1 \pm \sqrt{\gamma})^2$.

## 3. Optimization in Kernel Regime: Better NTK Conditioning Accelerates GLU Convergence

In this section, we analyze why GLU variants exhibit faster optimization. Using the NTK framework, we show that GLU's gating mechanism improves this conditioning in a two-layer ReGLU model. Finally, we empirically verify that the same conditioning improvement persists across activations and real datasets.

### 3.1. ReGLU NTK has Smaller Condition Number

Having established the connection between NTK conditioning and convergence rate (Sec.2.2), we now analyze how the GLU gating mechanism affects the NTK spectrum. Our goal in this subsection is to intuitively derive a relationship between the NTKs of two-layer ReLU and ReGLU models and to show that the latter yields a substantially smaller condition number.

**Problem setup.** We consider a two-layer neural network with input $\mathbf{x} \in \mathbb{R}^d$ and hidden width $m$. The non-GLU model is defined as

$$z(\mathbf{x}) = \mathbf{V}\,\phi(\mathbf{W}\mathbf{x}),$$

while the corresponding GLU model is

$$z(\mathbf{x}) = \mathbf{V}\big[(\mathbf{Px}) \odot \phi(\mathbf{Wx})\big],$$

where $\mathbf{W}, \mathbf{P} \in \mathbb{R}^{m\times d}$ and $\mathbf{V} \in \mathbb{R}^{1\times m}$ are learnable parameters. All parameters are initialized independently with zero-mean Gaussian distributions:

$$W_{ij} \sim \mathcal{N}(0, \sigma_w^2), \quad P_{ij} \sim \mathcal{N}(0, \sigma_p^2), \quad V_{ij} \sim \mathcal{N}(0, \sigma_v^2).$$

We note that in practice, to ensure the same amount of parameter, one typically scales down $m$ for GLU implementation. Here $m$ does not appear in the final result of condition number, hence to maintain clarity and align with real world scenario, we do not perform such scaling.

**Deriving the NTK matrix.** We compute the (expected) NTK for each model by taking expectations over the parameter distribution, according to law of large numbers. For the non-GLU mode,

$$
\begin{aligned}
K_{ij} =& \mathbb{E}_{\boldsymbol{\theta}} \langle \nabla_{\boldsymbol{\theta}} z(\mathbf{x}_i), \nabla_{\boldsymbol{\theta}} z(\mathbf{x}_j)\rangle \\
=& m\left[ \mathbb{E}_{\mathbf{w}}[\phi(\mathbf{w}^\top \mathbf{x}_i)\phi(\mathbf{w}^\top \mathbf{x}_j)] \right. \\
& \left. + \sigma_v^2 \mathbb{E}_{\mathbf{w}}[\phi'(\mathbf{w}^\top \mathbf{x}_i)\phi'(\mathbf{w}^\top \mathbf{x}_j)](\mathbf{x}_i^\top \mathbf{x}_j) \right],
\end{aligned}
\tag{3}
$$

where $\phi'$ denotes the derivative of $\phi$.

Similarly, for two-layer GLU model, we have[1]:

$$
\begin{aligned}
\tilde{K}_{ij} =& m\left[ (\sigma_v^2 + \sigma_p^2)\mathbb{E}_{\mathbf{w}}[\phi(\mathbf{w}^\top \mathbf{x}_i)\phi(\mathbf{w}^\top \mathbf{x}_j)](\mathbf{x}_i^\top \mathbf{x}_j) \right. \\
& \left. + \sigma_v^2 \sigma_p^2 \mathbb{E}_{\mathbf{w}}[\phi'(\mathbf{w}^\top \mathbf{x}_i)\phi'(\mathbf{w}^\top \mathbf{x}_j)](\mathbf{x}_i^\top \mathbf{x}_j)^2 \right],
\end{aligned}
\tag{4}
$$

Under LeCun initialization ($\sigma_w^2 = \sigma_p^2 = 1/d$ and $\sigma_v^2 = 1/m$), and using $\sigma_v^2 + \sigma_p^2 \approx \sigma_p^2$ for large hidden width $m$, we obtain the approximation

$$\tilde{K}_{ij} \approx K_{ij}\left(\sigma_p^2 \mathbf{x}_i^\top \mathbf{x}_j\right) = K_{ij}\left(\frac{\mathbf{x}_i^\top \mathbf{x}_j}{d}\right).$$

Equivalently, in matrix form,

$$\tilde{\mathbf{K}} \approx \mathbf{K} \odot \left(\frac{\mathbf{XX}^\top}{d}\right). \tag{5}$$

The relation in Eq. (5) reveals the key structural effect of the GLU gate: **the GLU NTK is obtained by Hadamard-multiplying the non-LU NTK with the data Gram matrix $\mathbf{XX}^\top/d$.** This operation makes the NTK matrix better conditioned, as will be seen in the following theorem.

**Theorem 3.1** (informal). *Consider two-layer ReLU and ReGLU models under LeCun initialization. Suppose the inputs $\{\mathbf{x}_i\}_{i=1}^n$ are i.i.d. standard Gaussian vectors, and*

---

$d + 1 < n$. *Then for their NTK matrices, consider their limiting spectral distribution, we have that*

*1) The largest eigenvalues scale as*

$$\lambda_{\max}(\mathbf{K}) = O\left(\frac{mn}{d}\right), \quad \lambda_{\max}(\tilde{\mathbf{K}}) = O\left(\frac{mn}{d^2}\right).$$

*2) The smallest eigenvalues both scale as $O(m)$, meanwhile*

$$\lambda_{\min}(\tilde{\mathbf{K}}) \geq \lambda_{\min}(\mathbf{K}).$$

*3) Consequently, the NTK condition numbers scales as*

$$\kappa(\mathbf{K}) = O\left(\frac{n}{d}\right), \quad \kappa(\tilde{\mathbf{K}}) = O\left(\frac{n}{d^2}\right).$$

This theorem reveals that the NTK spectrum of ReGLU is more "contracted" than ReLU: ReGLU model has smaller $\lambda_{\max}$ and larger $\lambda_{\min}$. This immediately leads to a better conditioned NTK matrix, which thereby accelerates the training convergence.

### 3.2. Proof Sketch of Theorem 3.1

Below we provide a proof sketch for Thm.3.1. The proof is completed in two steps: we first rewrite the NTK matrix as a sum of different components, then analyze their condition numbers.

We note that the argument below is simplified and aimed at conveying the main idea. The corresponding rigorous proof and precise asymptotics are provided in App.B.

**1) Rewriting the NTK matrix.**

Since we are considering ReLU activation, we can obtain the analytic form of (3) and (4) using results from arc-cosine kernel (Cho & Saul, 2010).

For ReLU kernel (3), using LeCun initialization, we have:

$$
\begin{aligned}
K_{ij} =& \left(\frac{1}{4} + \frac{\arcsin(\rho_{ij})}{2\pi}\right)\rho_{ij}\|\mathbf{x}_i\|\|\mathbf{x}_j\| \\
& + \frac{m\|\mathbf{x}_i\|\|\mathbf{x}_j\|}{2\pi d}\left(\sqrt{1 - \rho_{ij}^2} + (\pi - \arccos(\rho_{ij}))\rho_{ij}\right),
\end{aligned}
$$

where $\rho_{ij} = \frac{\mathbf{x}_i^\top \mathbf{x}_j}{\|\mathbf{x}_i\|\|\mathbf{x}_j\|}$ is the cosine similarity between samples $\mathbf{x}_i$ and $\mathbf{x}_j$.

Furthermore, since $\rho_{ij}$ is relatively small when $i \neq j$ for high dimensional inputs, we can well use Taylor expansion and keeping the first order of $\rho_{ij}$:

$$
K_{ij} \approx \begin{cases} \left(\dfrac{m}{2d} + \dfrac{1}{2}\right)\|\mathbf{x}_i\|^2, & i = j; \\[2ex] \left[\dfrac{m}{2\pi d} + \left(\dfrac{1}{4} + \dfrac{m}{4d}\right)\rho_{ij}\right]\|\mathbf{x}_i\|\|\mathbf{x}_j\|, & i \neq j. \end{cases}
$$

---

[1] Unless otherwise specified, quantities with a tilde ( ˜ ) refer to the GLU model.

Equivalently, we have the following matrix form:

$$\mathbf{K} = \underbrace{\alpha \mathbf{X}\mathbf{X}^\top}_{\text{data Gram matrix}} + \underbrace{\beta \mathbf{r}\mathbf{r}^\top}_{\text{rank-1 update}} + \underbrace{\gamma \mathbf{D}}_{\text{diagonal rectification}}, \quad (6)$$

with coefficients $\alpha = \frac{1}{4} + \frac{m}{4d}, \beta = \frac{m}{2\pi d}$ and $\gamma = \frac{1}{4} + \frac{m}{4d} - \frac{m}{2\pi d}$, all of which are positive with order $O(m/d)$. Here $\mathbf{r} \in \mathbb{R}^n$ is a vector with its $i$-th element $r_i = \|\mathbf{x}_i\|$, $\mathbf{D}$ is a diagonal matrix defined as $\mathbf{D} = \text{diag}\{r_1^2, \ldots, r_n^2\} = \text{diag}(\mathbf{X}\mathbf{X}^\top)$.

For ReGLU model, substituting (6) to (5), we obtain that

$$\tilde{\mathbf{K}} = \frac{\alpha}{d}(\mathbf{X}\mathbf{X}^\top) \odot (\mathbf{X}\mathbf{X}^\top) + \frac{\beta}{d}(\mathbf{r}\mathbf{r}^\top) \odot (\mathbf{X}\mathbf{X}^\top) + \frac{\gamma}{d}\mathbf{D}^2. \quad (7)$$

**2) Analyzing the condition number under Gaussian input assumption.**

Note that, we assume the inputs are i.i.d. standard Gaussian vectors, which give rise to the following three observations:

1. According to the definition (2), the data Gram matrix in (6) is actually a Wishart matrix, which is defined as $\mathbf{W} = \frac{1}{d}\mathbf{X}\mathbf{X}^\top$ with $X_{ij} \overset{\text{i.i.d.}}{\sim} \mathcal{N}(0,1), \mathbf{X} \in \mathbb{R}^{n \times d}$;

2. $\mathbf{D}$ is a diagonal matrix with $D_{ii} = \|\mathbf{x}_i\|^2$. Since we are studying the limiting spectral distribution, we have that $\mu(\mathbf{D}) = \mu(d\mathbf{I})$, where $\mu(\cdot) = \mathbb{E}\hat{\mu}(\cdot)$ stands for the expected empirical eigenvalue distribution;

3. Similarly, $\mu(\mathbf{r}\mathbf{r}^\top) = \mu(d\mathbf{1}\mathbf{1}^\top)$.

Hence we may further write the limiting spectral distribution of the NTK matrices as:

$$\mu(\mathbf{K}) \approx \mu(\alpha \mathbf{X}\mathbf{X}^\top + \beta d \mathbf{1}\mathbf{1}^\top + \gamma d \mathbf{I}) \quad \textbf{(non-GLU)},$$

$$\mu(\tilde{\mathbf{K}}) \approx \mu\left(\frac{\alpha}{d}(\mathbf{X}\mathbf{X}^\top) \odot (\mathbf{X}\mathbf{X}^\top) + \beta \mathbf{X}\mathbf{X}^\top + \gamma d \mathbf{I}\right) \quad \textbf{(GLU)}.$$

We now analyze the eigenspectrum of the matrix components respectively. Without loss of generality, we assume the eigenvalues are arranged in descending order: $\lambda_1 \geq \cdots \geq \lambda_n$. Note that $\lambda_i(\mathbf{K}) = \lambda_i(\mathbf{K} - \gamma d\mathbf{I}) + \gamma d$, hence we focus on the three components: **the Wishart matrix $\mathbf{W} = \frac{1}{d}\mathbf{X}\mathbf{X}^\top$, its self Hadamard product $\mathbf{W} \odot \mathbf{W}$, and rank-1 update $\mathbf{1}\mathbf{1}^\top$.**

For Wishart matrix, its eigenspectrum follows Marchenko-Pastur distribution (Marčenko & Pastur, 1967), we have:

$$\lambda_1(\mathbf{W}) = (1 + \sqrt{n/d})^2 \sim O(1 + n/d),$$

$$\lambda_n(\mathbf{W}) = \left(\max\{0, 1 - \sqrt{n/d}\}\right)^2 \overset{(n > d+1)}{=} 0.$$

For the self Hadamard product of Wishart matrix, $\mathbf{W} \odot \mathbf{W}$. The associated kernel mapping function is $f(x) = x^2$.

*Table 1.* Scaling of the extreme eigenvalues for the Wishart matrix, its self Hadamard product, and the rank-1 matrix.

| MATRIX | $\lambda_{\max}$ | $\lambda_{\min}$ |
|---|---|---|
| $\mathbf{X}\mathbf{X}^\top$ | $O(d+n)$ | $0$ |
| $\frac{1}{d}(\mathbf{X}\mathbf{X}^\top) \odot (\mathbf{X}\mathbf{X}^\top)$ | $O(d+n)$ | $O(d)$ |
| $d\mathbf{1}\mathbf{1}^\top$ | $dn$ | $0$ |

Therefore, we use the canonical result on kernel random matrix established by El Karoui (2010, Theorem 2.1), which states that, under mild regularity conditions, an inner product kernel random matrix can be approximated as

$$\mathbf{K} = \left(f(0) + f''(0)\frac{\text{Tr}(\mathbf{\Sigma}^2)}{2d^2}\right)\mathbf{1}\mathbf{1}^\top + f'(0)\frac{\mathbf{X}\mathbf{X}^\top}{d} + v_d \mathbf{I},$$

where

$$v_d = f\left(\frac{\text{Tr}(\mathbf{\Sigma})}{d}\right) - f(0) - f'(0)\frac{\text{Tr}(\mathbf{\Sigma})}{d}.$$

Here $\mathbf{\Sigma}$ is the population convariance matrix. In our setting $\mathbf{\Sigma} = \mathbf{I}$, which yields,

$$\mathbf{W} \odot \mathbf{W} \approx \frac{1}{d}\mathbf{1}\mathbf{1}^\top + \mathbf{I}.$$

Consequently, the extreme eigenvalues of $\mathbf{W} \odot \mathbf{W}$ satisfy

$$\lambda_1(\mathbf{W} \odot \mathbf{W}) \approx 1 + \frac{n}{d}, \quad \lambda_n(\mathbf{W} \odot \mathbf{W}) \approx 1.$$

For the rank-1 update matrix $\mathbf{1}\mathbf{1}^\top$, we have

$$\lambda_1\left(\mathbf{1}\mathbf{1}^\top\right) = n, \ \lambda_2\left(\mathbf{1}\mathbf{1}^\top\right) = \cdots = \lambda_n\left(\mathbf{1}\mathbf{1}^\top\right) = 0.$$

Tab.1 summarizes the results.

Now we can put them together. For symmetric matrices $\mathbf{P}$ and $\mathbf{Q}$, for any $k$, Weyl's inequality states that (Horn & Johnson, 2012, Theorem 4.3.1)

$$\lambda_k(\mathbf{P} + \mathbf{Q}) - \lambda_k(\mathbf{P}) \in [\lambda_n(\mathbf{Q}), \lambda_1(\mathbf{Q})].$$

Consider the largest eigenvalue of NTK matrix. According to Weyl's inequality, we have

$$\beta\lambda_1\left(d\mathbf{1}\mathbf{1}^\top\right) + \alpha\lambda_n\left(\mathbf{X}\mathbf{X}^\top\right) + \gamma d$$
$$\leq \lambda_1(\mathbf{K})$$
$$\leq \beta\lambda_1\left(d\mathbf{1}\mathbf{1}^\top\right) + \alpha\lambda_1\left(\mathbf{X}\mathbf{X}^\top\right) + \gamma d,$$

and

$$\alpha\lambda_1\left(\frac{1}{d}(\mathbf{X}\mathbf{X}^\top) \odot (\mathbf{X}\mathbf{X}^\top)\right) + \beta\lambda_n\left(\mathbf{X}\mathbf{X}^\top\right) + \gamma d$$
$$\leq \lambda_1(\tilde{\mathbf{K}})$$
$$\leq \alpha\lambda_1\left(\frac{1}{d}(\mathbf{X}\mathbf{X}^\top) \odot (\mathbf{X}\mathbf{X}^\top)\right) + \beta\lambda_1\left(\mathbf{X}\mathbf{X}^\top\right) + \gamma d.$$

Recall that $\alpha, \beta, \gamma \sim O(m/d)$. Therefore,

$$\lambda_1(\mathbf{K}) \sim O\left(\frac{mn}{d}\right), \lambda_1(\tilde{\mathbf{K}}) \sim O\left(\frac{mn}{d^2}\right).$$

Consider the smallest eigenvalue of NTK matrix. On one hand, rank-1 update does not affect the minimum eigenvalue of Wishart matrix, because according to Horn & Johnson (2012, Corollary 4.3.9),

$$\lambda_n(\alpha \mathbf{X}\mathbf{X}^\top) \leq \lambda_n(\alpha \mathbf{X}\mathbf{X}^\top + \beta d \mathbf{1}\mathbf{1}^\top) \leq \lambda_{n-1}(\alpha \mathbf{X}\mathbf{X}^\top),$$

and $\lambda_n(\alpha \mathbf{X}\mathbf{X}^\top) = \lambda_{n-1}(\alpha \mathbf{X}\mathbf{X}^\top) = 0$ since $n > d + 1$. Hence $\lambda_n(\mathbf{K}) = \gamma d \sim O(m)$. On the other hand, by Weyl's inequality,

$$\lambda_n(\tilde{\mathbf{K}}) \geq \lambda_n\left(\frac{\alpha}{d}(\mathbf{X}\mathbf{X}^\top) \odot (\mathbf{X}\mathbf{X}^\top)\right) + \gamma d \geq \lambda_n(\mathbf{K}).$$

Therefore, both $\lambda_n(\mathbf{K})$ and $\lambda_n(\tilde{\mathbf{K}})$ scale by $\sim O(m)$, and $\lambda_n(\tilde{\mathbf{K}}) > \lambda_n(\mathbf{K})$.

Since $\kappa = \frac{\lambda_1}{\lambda_n}$, we immediately obtain the final result.

### 3.3. Experimental Verification

To verify the correctness of our spectral approximations, we compare the estimated NTK condition numbers and extreme eigenvalues derived from Prop.B.6 and Prop.B.9 to numerical results.

Specifically, for numerical results, we construct the NTK matrices given in Prop.B.1 for both ReLU and ReGLU models, and compute their condition numbers by numerically evaluating the extreme eigenvalues using a symmetric eigensolver. For theoretical estimation, we use the approximation expressions of largest eigenvalues (Prop.B.6) and smallest eigenvalues (Prop.B.9) given in appendix. Note that for the largest eigenvalue of ReGLU model in Prop.B.6, we use the lower bound, which is itself a good approximation of the true value with negligible high order error terms.

As shown in Fig.2, the theoretical estimates closely match the numerically computed condition numbers across input dimensions, validating the correctness of our approximation.

Finally, we also introduce GLU in FFN block of ViT (Dosovitskiy et al., 2021) and GPT-2 (Radford et al., 2019). As illustrated in Fig.3, replacing the FFN blocks with GLU variants leads to a decrease in the condition number, providing strong empirical support for our theoretical analysis.

### 3.4. An Intuitive View: GLU Induces Diagonal Dominance in the NTK

To visualize the structural difference between the NTKs induced by GLU and non-GLU architectures, we plot heatmaps of the corresponding NTK matrices in Fig.4.

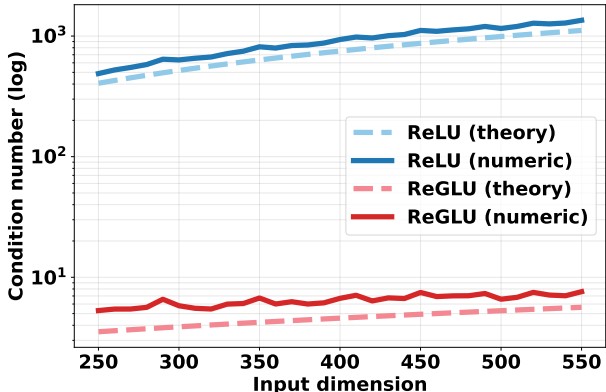

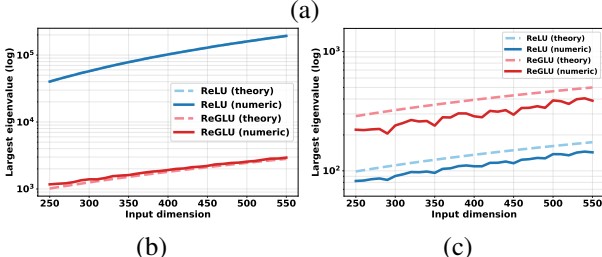

*Figure 2.* Comparison between theoretical and numerical NTK (a) condition numbers, (b) largest eigenvalues and (c) smallest eigenvalues for ReLU and ReGLU models. The theoretical predictions closely track the numerically computed ones.

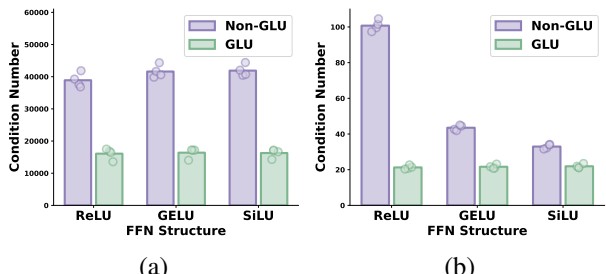

*Figure 3.* Condition number of (a) ViT and (b) GPT-2 under different activation choices.

Eq.(5) reveals that the GLU NTK can be viewed as a Hadamard reweighting of the non-GLU NTK by the data Gram matrix $\mathbf{X}\mathbf{X}^\top/d$. As is shown in Fig.4, **this gating mechanism significantly enhances the diagonal dominance of the kernel: diagonal entries become more pronounced, while off-diagonal entries are suppressed**.

In fact, by definition, the off-diagonal entries of NTK matrix denotes the model gradient correlation between different samples: $K_{ij} = \langle \nabla_{\boldsymbol{\theta}} z(\mathbf{x}_i), \nabla_{\boldsymbol{\theta}} z(\mathbf{x}_j) \rangle$. Let $\phi$ denote the model gradient angle, we have $\cos \phi_{ij} = \frac{K_{ij}}{\sqrt{K_{ii}K_{jj}}}$. Then according to Eq.(5), simple calculation gives:

$$\cos \tilde{\phi}_{ij} = \cos \phi_{ij} \cdot \cos \alpha_{ij}, \tag{8}$$

where $\alpha_{ij}$ represents the angle between sample $\mathbf{x}_i$ and $\mathbf{x}_j$. Consider similar samples where $\phi \leq \frac{\pi}{2}$, we immediately

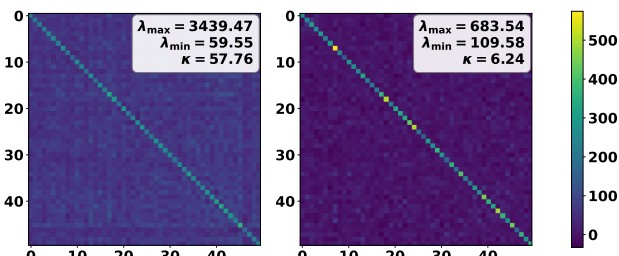

*Figure 4.* Visualization of the NTK matrices for ReLU (left) and ReGLU (right) models.

have that $\tilde{\phi} \geq \phi$. This result suggests that GLU structure can better seperate model gradients of different samples. According to Liu et al. (2025), better separated model gradients lead to faster convergence, which explains the reason why introducing GLU structure accelerates convergence.

## 4. Training Dynamics in the Kernel Regime: From NTK Spectrum to Loss Crossing

In previous section, we showed that GLU induces a more diagonally dominant NTK, leading to a smaller maximal eigenvalue and a larger minimal eigenvalue, and hence a more contracted spectrum. In this section, we show how this spectral reshaping leads to different convergence behaviors across training stages and explains the loss-crossing phenomenon in the kernel regime.

### 4.1. Direction-wise Dynamics and Geometric Intuition

We begin by recalling a basic property of gradient descent in the kernel regime. Let $\mathbf{e}_t = z_t(\mathbf{X}) - \mathbf{Y}$ denote the prediction error vector at iteration $t$, and let $(\lambda_i, \mathbf{v}_i)$ be the eigenpairs of the NTK matrix $\mathbf{K}$. For the mean squared error (MSE) loss, the error component along each eigendirection evolves as (Jacot et al., 2018; Lee et al., 2019)

$$\langle \mathbf{e}_{t+1}, \mathbf{v}_i \rangle = (1 - \eta \lambda_i)\langle \mathbf{e}_t, \mathbf{v}_i \rangle, \tag{9}$$

where $\eta$ is the learning rate.

Equation (9) implies that training proceeds in a direction-wise manner: **components aligned with larger eigenvalues decay rapidly, while components associated with smaller eigenvalues decay more slowly and dominate the late stages of optimization.** As a result, the overall convergence behavior is inherently stage-dependent, with different parts of the NTK spectrum governing different phases of training.

We can visualize this direction-wise behavior with a two-sample toy model ($n = 2$), where the NTK reduces to a $2 \times 2$ positive semidefinite matrix with two eigendirections.

The training trajectory can be visualized together with an ellipse centered at the optimal value $\mathbf{y}$, whose principal axes align with the NTK eigenvectors and whose axis lengths are proportional to $1/\lambda_i$.

The training tranjectory is plotted in Fig.5. As is analyzed in previous section, the ReLU NTK typically exhibits a larger $\lambda_{\max}$, leading to faster convergence along the top eigendirection at early stages of training (Fig.5(a, b)). But as training proceeds, convergence becomes increasingly constrained by the direction associated with the smallest eigenvalue. Since ReGLU exhibits a larger $\lambda_{\min}$, it decreases the remaining error more effectively in this late stage, and eventually overtakes ReLU (Fig.5(c)). Finally, both models converge toward target, while ReGLU model takes a more "direct" trajectory (Fig.5(d)).

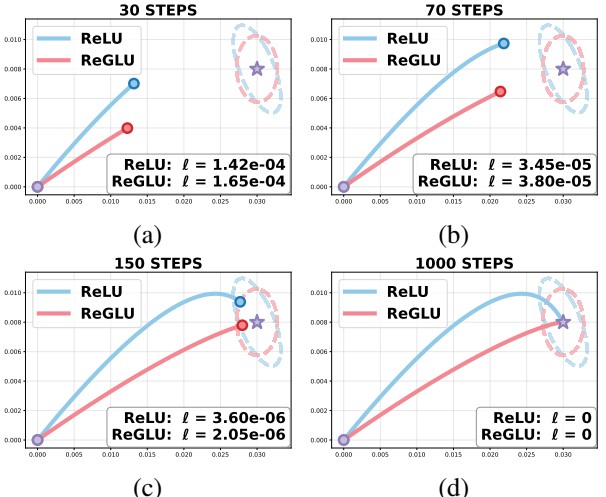

*Figure 5.* Training trajectories in a two-sample toy model for ReLU and ReGLU. (a) 30 steps. (b) 70 steps. (c) 150 steps. (d) 1000 steps.

### 4.2. Loss-Crossing Phenomenon

We now formalize the above intuition and connect it to the loss-crossing phenomenon. Our analysis is in the infinite-width limit ($m \to \infty$), where the model operates in the kernel regime, and the expected training loss admits a closed-form expression.

**Proposition 4.1.** *Consider the case when $m \to \infty$, i.e., kernel regime. Denote by $\mathbf{K}$ the NTK matrix, then the expected loss of the two-layer MLP evolves as*

$$\mathbb{E}_{\boldsymbol{\theta}}[L_k] \propto \mathrm{Tr}[(\mathbf{I} - \eta \mathbf{K})^{2k}\mathbf{K}] + \mathbf{Y}^\top (\mathbf{I} - \eta \mathbf{K})^{2k}\mathbf{Y}.$$

The above expression captures the expected loss evolution of our two-layer model in the kernel regime. As a corollary, we can provide a finer-grained analysis of training dynamics of the two-layer ReLU/ReGLU MLP models.

**Corollary 4.2.** *Consider two-layer ReLU/ReGLU MLP models with gaussian inputs* $\mathbf{x}_i \sim \mathcal{N}(\mathbf{0}, \mathbf{I}) \in \mathbb{R}^d$, *we have that:*

*1) At the early stage when* $(\eta k)$ *is small, as long as* $\mathbf{Y}^\top(\mathbf{K} - \tilde{\mathbf{K}})\mathbf{Y} \geq 0$, $d \geq 5$ *and* $n \geq 300$, *it holds that* $\mathbb{E}_{\boldsymbol{\theta}}[L_k - \tilde{L}_k] < 0$. *That is, ReLU model converges faster than ReGLU model.*

*2) At the late stage when the minimum eigenvalue* $\lambda_{\min}$ *dominates the training process, for sufficiently large* $k$, *it holds that* $\mathbb{E}_{\boldsymbol{\theta}}[L_k - \tilde{L}_k] > 0$. *That is, ReGLU model takes over and converges faster than ReLU model.*

The proofs of Prop.4.1 and Cor.4.2 are deferred to App.C. This corollary indicates a two phase convergence behavior, where ReLU model converges faster than ReGLU model at the initial stage, but is overtaken later by the ReGLU model. As shown in Fig.6, this theoretical finding is consistent with empirical observations.

Furthermore, this behavior is consistently observed across different activation pairs (GELU/GEGLU and SiLU/SwiGLU), as well as on both synthetic Gaussian data and MNIST. We further find that the prominence of loss crossing depends on the learning rate: increasing the learning rate uniformly accelerates convergence across eigendirections, thereby diminishing the early-stage advantage of non-GLU models. This trend is consistent with the NTK dynamics in Eq.(9) and supports the spectral interpretation.

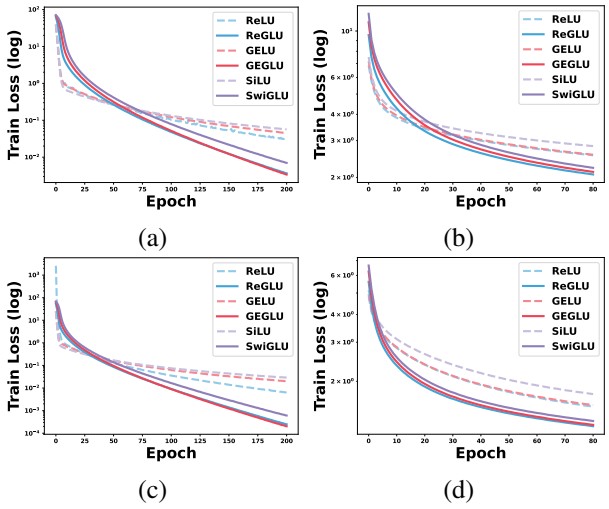

(a)   (b)

(c)   (d)

*Figure 6.* Training loss curves on two-layer MLP models. (a) Gaussian data with learning rate 0.005. (b) MNIST with learning rate $1 \times 10^{-5}$. (c) Gaussian data with learning rate 0.008. (d) MNIST with learning rate $5 \times 10^{-5}$.

## 5. Generalization Gap Analysis

Finally, we examine whether GLU variants help reduce the generalization gap. Intuitively, the multiplicative gating in

GLU introduces second-order nonlinearity, which may enhance expressiveness and reduce generalization gap. However, our empirical results suggest that this effect is limited. Fig.7 plots the generalization gap $L_\mathcal{D} - L_S$ against the training loss $L_S$ for models with and without GLU. **Additional results are provided in App.D.**

We observe that GLU-based models and non-GLU models exhibit highly overlapping distributions in the $(L_S, L_\mathcal{D} - L_S)$ plane. For a given training loss level, the generalization gap achieved by GLU and non-GLU models is nearly indistinguishable. This suggests that, **in contrast to its clear impact on optimization dynamics, GLU provides limited advantage in reducing the generalization gap**. In comparison, the choice of optimizer has a different impact on generalization gap. In Fig.7, we observe that switching from SGD to AdamW shifts the scatter downward, leading to a smaller generalization gap at comparable training losses for both ReLU and ReGLU models. This effect is significantly stronger than the differences induced by the presence or absence of GLU.

Furthermore, we also train GPT-2 on FineWeb-Edu dataset with SiLU and SwiGLU activation, since most modern open-source LLMs use SwiGLU as the FFN architecture. The results are shown in Fig.7(b). From the figure, we can also see that introducing GLU does not provide apparent advantage in reducing generalization gap.

To rigorously quantify this observation, we further perform a two-sample permutation test based on Energy Distance to compare the joint distributions of $(L_S, L_\mathcal{D} - L_S)$ for models with and without GLU. The statistical test fails to find evidence of a significant shift in the generalization gap distribution ($p \geq 0.05$), supporting our observation that GLU's impact on this specific metric is marginal.

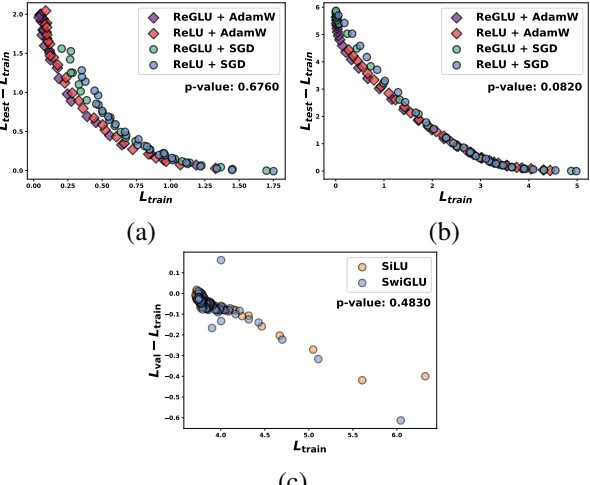

(a)   (b)

(c)

*Figure 7.* Generalization gap vs training loss. (a) MLP Mixer on CIFAR-10. (b) ViT on Tiny ImageNet. (c) GPT-2 on FineWeb-Edu.

## 6. Conclusion

In this work, we provide a theoretical investigation of the advantage of GLU structure. By studying GLU in a two-layer model and analyzing their behavior in the kernel regime, we show that such multiplicative gating leads to an improved spectral conditioning of the NTK and yields faster optimization dynamics. Based on our theoretical result, we clarify how GLU reshapes convergence behavior during training. In contrast, our empirical results suggest that GLU has a limited impact on the generalization gap under the considered settings, indicating that its primary benefit lies in accelerating training convergence rather than reducing the generalization gap. Finally, we believe that further understanding whether and how multiplicative gating can influence training and learning behavior in broader regimes remains an interesting direction for future work.

## Acknowledgements

This work was supported in part by National Natural Science Foundation of China: 62525212, U23B2051, 62236008, 62441232, 62521007, 62576332, and U21B2038, in part by Youth Innovation Promotion Association CAS, in part by the Strategic Priority Research Program of the Chinese Academy of Sciences, Grant No. XDB0680201, in part by the project ZR2025ZD01 supported by Shandong Provincial Natural Science Foundatio, in part by the China National Postdoctoral Program for Innovative Talents under Grant BX20250377, in part by the Beijing Major Science and Technology Project under Contract No. Z251100008125059, and in part by Beijing Academy of Artificial Intelligence (BAAI).

## Impact Statement

This paper presents work whose goal is to advance the field of Machine Learning. There are many potential societal consequences of our work, none which we feel must be specifically highlighted here.

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

# Appendix

## Contents

# A. Related Works

This paper aims to explain the optimization advantages of GLU variants from the perspective of the kernel regime. Accordingly, we review three closely related lines of work. We first summarize prior studies on gated mechanisms and GLU variants. We then review relevant results on neural tangent kernels (NTK) that connect kernel spectra with optimization dynamics. Finally, we discuss works in random matrix theory, which provide the main analytical tools used in this paper.

## A.1. Gated Linear Unit (GLU) and its variants

Gating mechanisms have long been employed in neural networks to improve optimization and model expressivity (Srivastava et al., 2015; Jozefowicz et al., 2016; Dey & Salem, 2017). Their early success can be traced back to LSTM architectures (Hochreiter & Schmidhuber, 1997). More recently, gating ideas have been extended to large-scale models, such as gated attention mechanism in large language models, which demonstrates consistent empirical improvements (Qiu et al., 2025).

Gated Linear Units (GLU) can be viewed as a concrete instantiation of gating mechanisms within feed-forward networks (FFN). GLU were first introduced by Dauphin et al. (2017), where a multiplicative gating structure was applied after convolutional layers to improve gradient propagation. Subsequently, Shazeer (2020) proposed several GLU variants, including ReGLU, GEGLU, and SwiGLU, and showed that replacing the standard FFN activation in Transformers with these gated variants leads to systematic performance gains, with SwiGLU being particularly effective. As a result, SwiGLU has become a standard architectural component in many modern open-source large language models, such as Llama 3 (Grattafiori et al., 2024), DeepSeek-V2 (Liu et al., 2024), Qwen 3 (Yang et al., 2025), and Kimi K2 (Team et al., 2025).

Despite their widespread empirical success, the theoretical reasons behind the effectiveness of GLU variants remain poorly understood. In their original work, Shazeer (2020) attributed the observed improvements to what they described as "divine benevolence", underscoring the absence of a principled explanation. More recently, Zhong et al. (2025) challenged the common intuition that gating in FFN acts as a simple forgetting mechanism, and emphasized the lack of theoretical understanding. In particular, how GLU and its variants influence optimization dynamics from a kernel perspective remains largely unexplored, which motivates the focus of this paper.

## A.2. Neural Tangent Kernel (NTK)

Since the introduction of the neural tangent kernel (NTK) for infinitely wide neural networks (Jacot et al., 2018; Chizat et al., 2019), its spectral properties have been extensively studied (Lee et al., 2019; Fan & Wang, 2020; Chen et al., 2022). Prior works have shown that the eigenvalue distribution of NTK exhibits strong similarities to classical kernels such as the Laplace kernel (Geifman et al., 2020). More recently, Murray et al. (2023) characterized the NTK spectrum through power series expansions, deriving the scaling behavior of the smallest eigenvalue for two-layer ReLU networks in the infinite-width limit.

Beyond spectral characterization, similar to the Hessian whose spectrum directly implies the geometry of the loss landscape, the NTK spectrum has been shown to play a central role in training dynamics and convergence behavior of neural networks (Lyu et al., 2025; Allen-Zhu et al., 2019; Liu et al., 2020). In particular, de Ryck et al. (2024) demonstrated that better conditioning (i.e., smaller condition number) of the NTK leads to faster convergence of gradient-based training. Liu et al. (2025) further analyzed worst-case convergence rates and compared the conditioning properties of linear and ReLU-activated networks.

Together, these results shed light on the theoretical connection between NTK spectral properties and optimization efficiency, forming an important foundation for the analysis in this paper.

## A.3. Random Matrix Theory

Random matrix theory provides fundamental tools for analyzing the spectral properties of large random matrices (Feier, 2012). Classical results characterize the limiting spectral behavior of sample covariance and Wigner-type matrices (Hsu, 1939; Wigner, 1958). These ideas have been applied to kernel methods through the study of kernel random matrices. A seminal contribution by El Karoui (2010) established a linearization framework that enables the spectral analysis of kernel random matrices via approximated random matrix models. Subsequent works further investigated limiting spectral distributions (Do & Vu, 2013) and concentration properties of inner-product kernel matrices (Amini & Razaee, 2021), as well as their implications for NTK and generalization performance (Mei et al., 2022). More recently, Pandit et al. (2024) studied kernel random matrices in the quadratic regime and derived spectral results for self Hadamard products of Gram

matrices.

In this work, we draw on these results as technical tools for analyzing the spectral properties of NTK matrices induced by GLU structure.

## B. Approximating the NTK Condition Number

In this section, we present a rigorous proof of Thm.3.1. The proof is divided into three steps. First, we derive explicit expressions for the NTK matrices of two-layer ReLU and ReGLU models. Next, we establish analytic approximations for the largest eigenvalues. Finally, we derive corresponding approximations for the smallest eigenvalues.

Importantly, all results are obtained in explicit analytic form rather than relying on asymptotic arguments. Numerical experiments are further provided to verify the accuracy of these approximations, thereby supporting both their rigor and practical relevance.

### B.1. Derivation of the NTK Matrix

In this section, we offer the rigorous derivation of the ReLU NTK matrix (Eq.(6)) and its ReGLU conterpart. The results are summarized in the following lemma.

**Proposition B.1.** *Consider the two-layer ReLU and ReGLU models described in Sec.3.1. Denote by $\mathbf{X} \in \mathbb{R}^{n \times d}$ the sample matrix, $\mathbf{r} = [\|\mathbf{x}_1\|, \ldots, \|\mathbf{x}_n\|]$, and $\mathbf{D} = \mathrm{diag}\left\{ r_1^2, \ldots, r_n^2 \right\}$. Then,*

*1) For ReLU model, the $ij$-th element of its NTK matrix is approximately*

$$\mathbf{K} = \alpha \mathbf{X}\mathbf{X}^\top + \beta \mathbf{r}\mathbf{r}^\top + \gamma \mathbf{D}. \tag{10}$$

*Here $\alpha = \frac{1}{4} + \frac{m}{4d}, \beta = \frac{m}{2\pi d}, \gamma = \frac{1}{4} + \frac{m}{4d} - \frac{m}{2\pi d}$.*

*2) For ReGLU model, the $ij$-th element of its NTK matrix is approximately*

$$\tilde{\mathbf{K}} = \tilde{\alpha}(\mathbf{X}\mathbf{X}^\top) \odot (\mathbf{X}\mathbf{X}^\top) + \tilde{\beta}(\mathbf{r}\mathbf{r}^\top) \odot (\mathbf{X}\mathbf{X}^\top) + \tilde{\gamma}\mathbf{D}^2. \tag{11}$$

*Here $\tilde{\alpha} = \frac{m}{4d^2} + \frac{1}{2d}, \tilde{\beta} = \frac{1}{2\pi d} + \frac{m}{2\pi d^2}, \tilde{\gamma} = \frac{1}{2d} - \frac{1}{2\pi d} + \frac{m}{4d^2} - \frac{m}{2\pi d^2}$.*

*Proof.* The proof is carried out in two main steps. First, we derive a general expression for the NTK matrix that is independent of the specific activation function. Then, by specializing to the arc-cosine kernel and applying a Taylor expansion, we obtain the final forms for the ReLU and ReGLU models.

**1) Obtaining the general form of NTK matrix.**

Consider ReLU model first:

$$z(\mathbf{x}) = \mathbf{V}\phi(\mathbf{W}\mathbf{x}).$$

Taking derivative w.r.t all parameters, we get:

$$\frac{\partial z(\mathbf{x})}{\partial V_k} = \phi(\mathbf{W}_k^\top \mathbf{x}), \qquad \frac{\partial z(\mathbf{x})}{\partial W_{ks}} = V_k \phi'(\mathbf{W}_k^\top \mathbf{x}) x_s,$$

where $\mathbf{W}_k$ stands for the $k$-th row of $\mathbf{W}$.

Hence we have:

$$\langle \nabla_\mathbf{V} z(\mathbf{x}_i), \nabla_\mathbf{V} z(\mathbf{x}_j) \rangle = \sum_{k=1}^m \phi(\mathbf{W}_k^\top \mathbf{x}_i)\phi(\mathbf{W}_k^\top \mathbf{x}_j).$$

$$\langle \nabla_\mathbf{W} z(\mathbf{x}_i), \nabla_\mathbf{W} z(\mathbf{x}_j) \rangle = (\mathbf{x}_i^\top \mathbf{x}_j) \sum_{k=1}^m V_k^2 \phi'(\mathbf{W}_k^\top \mathbf{x}_i)\phi'(\mathbf{W}_k^\top \mathbf{x}_j).$$

Taking expectation, we obtain the (expected) NTK matrix:

$$\begin{aligned} K_{ij} &= \mathbb{E}_{\boldsymbol{\theta}}\left[ \langle \nabla_{\boldsymbol{\theta}} z(\mathbf{x}_i), \nabla_{\boldsymbol{\theta}} z(\mathbf{x}_j) \rangle \right] \\ &= m\left( \mathbb{E}_\mathbf{w}\left[ \phi(\mathbf{w}^\top \mathbf{x}_i)\phi(\mathbf{w}^\top \mathbf{x}_j) \right] + \sigma_v^2 \mathbb{E}_\mathbf{w}\left[ \phi'(\mathbf{w}^\top \mathbf{x}_i)\phi'(\mathbf{w}^\top \mathbf{x}_j) \right] (\mathbf{x}_i^\top \mathbf{x}_j) \right). \end{aligned} \tag{12}$$

Here $\mathbf{w} \sim \mathcal{N}(0, \mathbf{I})$ is the distribution of each row of $\mathbf{W}$.

For ReGLU model, which is defined as

$$z(\mathbf{x}) = \mathbf{V}\left[(\mathbf{Px}) \odot \phi(\mathbf{Wx})\right].$$

Similarly, we have:

$$\frac{\partial z(\mathbf{x})}{\partial V_k} = (\mathbf{P}_k^\top \mathbf{x})\phi(\mathbf{W}_k^\top \mathbf{x}), \quad \frac{\partial z(\mathbf{x})}{\partial P_{ks}} = V_k \phi(\mathbf{W}_k^\top \mathbf{x}) x_s, \quad \frac{\partial z(\mathbf{x})}{\partial W_{ks}} = V_k (\mathbf{P}_k^\top \mathbf{x})\phi'(\mathbf{W}_k^\top \mathbf{x}) x_s.$$

Hence

$$\langle \nabla_{\mathbf{V}} z(\mathbf{x}_i), \nabla_{\mathbf{V}} z(\mathbf{x}_j) \rangle = \sum_{k=1}^m (\mathbf{P}_k^\top \mathbf{x}_i)(\mathbf{P}_k^\top \mathbf{x}_j)\phi(\mathbf{W}_k^\top \mathbf{x}_i)\phi(\mathbf{W}_k^\top \mathbf{x}_j),$$

$$\langle \nabla_{\mathbf{P}} z(\mathbf{x}_i), \nabla_{\mathbf{P}} z(\mathbf{x}_j) \rangle = (\mathbf{x}_i^\top \mathbf{x}_j) \sum_{k=1}^m V_k^2 \phi(\mathbf{W}_k^\top \mathbf{x}_i)\phi(\mathbf{W}_k^\top \mathbf{x}_j),$$

$$\langle \nabla_{\mathbf{W}} z(\mathbf{x}_i), \nabla_{\mathbf{W}} z(\mathbf{x}_j) \rangle = (\mathbf{x}_i^\top \mathbf{x}_j) \sum_{k=1}^m V_k^2 (\mathbf{P}_k^\top \mathbf{x}_i)(\mathbf{P}_k^\top \mathbf{x}_j)\phi'(\mathbf{W}_k^\top \mathbf{x}_i)\phi'(\mathbf{W}_k^\top \mathbf{x}_j),$$

And eventually,

$$
\begin{aligned}
\tilde{K}_{ij} &= \mathbb{E}_{\boldsymbol{\theta}}[\langle \nabla_{\boldsymbol{\theta}} z(\mathbf{x}_i), \nabla_{\boldsymbol{\theta}} z(\mathbf{x}_j) \rangle] \\
&= m\left((\sigma_v^2 + \sigma_p^2)\mathbb{E}_{\mathbf{w}}[\phi(\mathbf{w}^\top \mathbf{x}_i)\phi(\mathbf{w}^\top \mathbf{x}_j)](\mathbf{x}_i^\top \mathbf{x}_j) + \sigma_v^2 \sigma_p^2 \mathbb{E}_{\mathbf{w}}[\phi'(\mathbf{w}^\top \mathbf{x}_i)\phi'(\mathbf{w}^\top \mathbf{x}_j)](\mathbf{x}_i^\top \mathbf{x}_j)^2\right).
\end{aligned}
\tag{13}
$$

## 2) Using arc-cosine kernel and Taylor approximation.

Since we are considering ReLU activated models, we can use arc-cosine kernel to get rid of the expectation factors. Specifically, we have (Cho & Saul, 2010),

$$\mathbb{E}_{\mathbf{w}}[\phi(\mathbf{w}^\top \mathbf{x}_i)\phi(\mathbf{w}^\top \mathbf{x}_j)] = \frac{\sigma_w^2 \|\mathbf{x}_i\|\|\mathbf{x}_j\|}{2\pi}\left(\sqrt{1 - \rho_{ij}^2} + \left(\pi - \arccos \rho_{ij}\right)\rho_{ij}\right),$$

$$\mathbb{E}_{\mathbf{w}}[\phi'(\mathbf{w}^\top \mathbf{x}_i)\phi'(\mathbf{w}^\top \mathbf{x}_j)] = \frac{1}{2\pi}\left(\pi - \arccos \rho_{ij}\right).$$

Here $\rho_{ij} := \frac{\mathbf{x}_i^\top \mathbf{x}_j}{\|\mathbf{x}_i\|\|\mathbf{x}_j\|}$ is the cosine similarity between samples $\mathbf{x}_i$ and $\mathbf{x}_j$.

Note that when $i \neq j$, for high dimensional inputs, $\rho_{ij} \to 0$. Hence we use Taylor expansion to approximate this term. We have:

$$\arcsin(z) = z + O(z^3), \quad \arccos(z) = \frac{\pi}{2} - \arcsin(z) = \frac{\pi}{2} - z + O(z^3), \quad \sqrt{1 - z^2} = 1 - \frac{z^2}{2} + O(z^3).$$

Substituting them back to (12) and (13), we obtain that:

1) For ReLU model, the $ij$-th element of its NTK matrix is approximately

$$
K_{ij} = \begin{cases}
\left(\dfrac{m}{2d} + \dfrac{1}{2}\right)\|\mathbf{x}_i\|^2, & i = j; \\
\left[\dfrac{m}{2\pi d} + \left(\dfrac{1}{4} + \dfrac{m}{4d}\right)\rho_{ij} + \left(\dfrac{1}{2\pi} + \dfrac{m}{4\pi d}\right)\rho_{ij}^2\right]\|\mathbf{x}_i\|\|\mathbf{x}_j\|, & i \neq j.
\end{cases}
\tag{14}
$$

2) For ReGLU model, the $ij$-th element of its NTK matrix is approximately

$$
\tilde{K}_{ij} = \begin{cases}
\left(\dfrac{m}{2d^2} + \dfrac{1}{d}\right)\|\mathbf{x}_i\|^4, & i = j; \\
\left[\left(\dfrac{1}{2\pi d} + \dfrac{m}{2\pi d^2}\right)\rho_{ij} + \left(\dfrac{1}{2d} + \dfrac{m}{4d^2}\right)\rho_{ij}^2\right]\|\mathbf{x}_i\|^2\|\mathbf{x}_j\|^2, & i \neq j.
\end{cases}
\tag{15}
$$

Finally, for both models, we only keep the terms where $\rho_{ij}$ can be absorbed into $\mathbf{x}_i^\top \mathbf{x}_j$. That is, we drop the $\rho_{ij}^2$ term in ReLU model but keep it in ReGLU model. Then we obtain the final result.

$\square$

## B.2. Estimating the Largest Eigenvalue

In this section, we seek to estimate the largest eigenvalue of the two-layer ReLU and ReGLU models. We begin by recalling the classical Weyl inequality (Horn & Johnson, 2012, Theorem 4.3.1).

**Theorem B.2** (Weyl's inequality). *Let $\mathbf{A}, \mathbf{B}$ be Hermitian matrices, with spectrum ordered in descending order $\lambda_1 \geq \cdots \geq \lambda_n$. Then,*

$$\lambda_{i+j-1}(\mathbf{A} + \mathbf{B}) \leq \lambda_i(\mathbf{A}) + \lambda_j(\mathbf{B}) \leq \lambda_{i+j-n}(\mathbf{A} + \mathbf{B}).$$

The following Corollary of Weyl's inequality is used throughout our analysis.

**Corollary B.3** (Spectral stability). *Let $\mathbf{A}, \mathbf{B}$ be Hermitian matrices, with spectrum ordered in descending order $\lambda_1 \geq \cdots \geq \lambda_n$. Then,*

$$\lambda_k(\mathbf{A} + \mathbf{B}) - \lambda_k(\mathbf{A}) \in [\lambda_n(\mathbf{B}), \lambda_1(\mathbf{B})],$$
$$|\lambda_k(\mathbf{A} + \mathbf{B}) - \lambda_k(\mathbf{A})| \leq \|\mathbf{B}\|_{op}.$$

The following lemma is also useful, which uses the maximum and minimum of row-sums to bound the largest eigenvalue of any non-negative symmetric matrix.

**Lemma B.4** (Row-sum bound for non-negative symmetric matrix). *Suppose $\mathbf{A} \in \mathbb{R}^{n \times n}$ is a non-negative real symmetric matrix, i.e., $A_{ij} \geq 0$. Then,*

$$\min_i \sum_{j=1}^{n} A_{ij} \leq \lambda_1(\mathbf{A}) \leq \max_i \sum_{j=1}^{n} A_{ij}.$$

*Proof.* The upper bound is from Gershgorin disc theorem. We have (Horn & Johnson, 2012, Corollary 6.1.5):

$$\rho(\mathbf{A}) \leq \min \left\{ \max_i \sum_{j=1}^{n} |A_{ij}|, \max_j \sum_{i=1}^{n} |A_{ij}| \right\}.$$

Here $\rho(\mathbf{A})$ is the spectral radius of $\mathbf{A}$, i.e., $\rho(\mathbf{A}) = \max \{|\lambda_1(\mathbf{A})|, \ldots, |\lambda_n(\mathbf{A})|\}$. Hence $\lambda_1(\mathbf{A}) \leq \rho(\mathbf{A}) \leq \max_i \sum_{j=1}^{n} A_{ij}$.

Now consider the lower bound. By property of Rayleigh quotient (Horn & Johnson, 2012, Theorem 4.2.2), we have:

$$\lambda_1(\mathbf{A}) = \max_{\mathbf{x} \neq \mathbf{0}} \frac{\mathbf{x}^\top \mathbf{A} \mathbf{x}}{\mathbf{x}^\top \mathbf{x}} \geq \frac{1}{n} \mathbf{1}^\top \mathbf{A} \mathbf{1} = \frac{1}{n} \sum_{i=1}^{n} \left( \sum_{j=1}^{n} A_{ij} \right) \geq \min_i \sum_{j=1}^{n} A_{ij}.$$

$\square$

Finally, we will also use the BBP phase transition (Baik et al., 2005; Benaych-Georges & Nadakuditi, 2011):

**Theorem B.5** (Theorem 2.1, (Benaych-Georges & Nadakuditi, 2011), restated). *Let $\mathbf{X}_n$ be an $n \times n$ symmetric random matrix with largest eigenvalue $\lambda_1(\mathbf{X}_n)$. Assume that the empirical spectral distribution $\mu_{\mathbf{X}_n} := \frac{1}{n} \sum_{j=1}^{n} \delta_{\lambda_j(\mathbf{X}_n)}$ converges almost surely, in the weak sense, to a non-random probability measure $\mu_{\mathbf{X}}$ supported by $a, b$, and that $\lambda_n(\mathbf{X}_n) \overset{a.s.}{\to} a, \lambda_1(\mathbf{X}_n) \overset{a.s.}{\to} b$. Define $\tilde{\mathbf{X}}_n = \mathbf{X}_n + \mathbf{P}_n$, where $\mathbf{P} = \theta \mathbf{u} \mathbf{u}^\top (\theta > 0)$ is a rank-1 signal update to $\mathbf{X}$. Let $G_{\mu_{\mathbf{X}}}$ denote the Cauchy transform of $\mu_{\mathbf{X}}$:*

$$G_{\mu_{\mathbf{X}}}(z) := \int \frac{1}{z - t} d\mu_{\mathbf{X}}(t), \qquad z \notin \operatorname{supp}(\mu_{\mathbf{X}}),$$

*and set*

$$G_{\mu_{\mathbf{X}}}(b^+) := \lim_{z \downarrow b} G_{\mu_{\mathbf{X}}}(z) \in (0, +\infty].$$

*Then the largest eigenvalue of $\tilde{\mathbf{X}}_n$ satisfies,*

$$\lambda_1(\tilde{\mathbf{X}}_n) \xrightarrow{a.s.} \begin{cases} G_{\mu_{\mathbf{X}}}^{-1}\left(\frac{1}{\theta}\right), & \text{if } \frac{1}{\theta} < G_{\mu_X}(b^+), \\ b, & \text{if } \frac{1}{\theta} \geq G_{\mu_X}(b^+), \end{cases}$$

*where $G_{\mu_{\mathbf{X}}}^{-1}$ denotes the inverse of $G_{\mu_{\mathbf{X}}}$.*

We are now prepared to prove the following lemma, which gives estimation of the largest eigenvalue of the NTK matrix.

**Proposition B.6** (Largest eigenvalue). *Consider the two-layer ReLU and ReGLU models described in Sec.3.1.*

*1) For ReLU model, the largest eigenvalue of its NTK matrix is given by*

$$\lambda_1(\mathbf{K}) \approx \frac{m}{2\pi} \cdot n + \frac{d}{2} + \frac{(\pi-1)m}{2\pi}.$$

*2) For ReGLU model, the largest eigenvalue of its NTK matrix is given by*

$$\left(\frac{m}{4d} + \frac{1}{2}\right) n + \frac{m}{2} - \frac{m}{2\pi} + d - \frac{d}{2\pi} \lesssim \lambda_1(\tilde{\mathbf{K}}) \lesssim \left(\frac{m}{4d} + \frac{m}{2\pi d} + \frac{1}{2} + \frac{1}{2\pi}\right) n + \frac{m}{2} + d.$$

*Therefore,*

$$\lambda_1(\mathbf{K}) = \Theta(mn), \qquad \lambda_1(\tilde{\mathbf{K}}) = \Theta(mn/d).$$

*Proof.* **1) ReLU model.**

For ReLU model, we note that by law of large numbers, the expression can be approximately written as:

$$\mathbf{K} \approx \alpha \mathbf{X}\mathbf{X}^\top + \beta d \mathbf{1}\mathbf{1}^\top + \gamma d\mathbf{I}.$$

This form of expression satisfies the condition of Thm.B.5. To see this, we define $\mathbf{W} = \frac{1}{d}\mathbf{X}\mathbf{X}^\top$ and $\mathbf{M} = \frac{\mathbf{K}}{d\alpha} - \frac{\gamma}{\alpha}\mathbf{I} = \mathbf{W} + \frac{\beta n}{\alpha}\mathbf{u}\mathbf{u}^\top$, where $\mathbf{u} = \frac{1}{\sqrt{n}}\mathbf{1}$ is a unit vector. Note that $\mathbf{W}$ is the Wishart matrix, the Cauchy tranformation of which has the following form:

$$G_{\mu_\mathbf{W}}(z) = \frac{-1 + c + z - \sqrt{(z-a)(z-b)}}{2cz},$$

and its inverse function:

$$G_{\mu_\mathbf{W}}^{-1}(y) = \frac{(c-1)y - 1}{y(cy-1)}.$$

where $c = \frac{n}{d}, a = (1-\sqrt{c})^2, b = (1+\sqrt{c})^2$. Hence

$$G_{\mu_\mathbf{W}}(b^+) = \lim_{z\downarrow b} G_{\mu_\mathbf{W}}(z) = (c+\sqrt{c})^{-1}$$

and

$$G_{\mu_\mathbf{W}}^{-1}\left(\frac{\alpha}{\beta n}\right) = \frac{\beta n}{\alpha} + 1 + \frac{\alpha}{\beta\sqrt{nd} - \alpha}.$$

Since $\mathbf{K} = \alpha d\mathbf{M} + \gamma d\mathbf{I}$, by Thm.B.5, we have that:

$$\lambda_1(\mathbf{K}) \approx \beta dn + (\alpha+\gamma)d = \frac{m}{2\pi} \cdot n + \frac{d}{2} + \frac{(\pi-1)m}{2\pi}$$

when $\frac{\beta n}{\alpha} > c + \sqrt{c}$, or $\left(\frac{\beta\sqrt{d}}{\alpha} - \frac{1}{\sqrt{d}}\right)\sqrt{n} > 1$, which is a quite minor condition since LHS is of order $\sqrt{nd}$.

**2) ReGLU model.**

For ReGLU model, similarly, we have:

$$\tilde{\mathbf{K}} \approx \tilde{\alpha}(\mathbf{X}\mathbf{X}^\top) \odot (\mathbf{X}\mathbf{X}^\top) + \tilde{\beta} d\mathbf{X}\mathbf{X}^\top + \tilde{\gamma} d^2\mathbf{I}.$$

We know that $\mathbf{X}\mathbf{X}^\top$ is positive semi-definite. Therefore, by Schur product theorem (Horn & Johnson, 2012, Theorem 7.5.3), $(\mathbf{X}\mathbf{X}^\top) \odot (\mathbf{X}\mathbf{X}^\top)$ is also positive semi-definite. Hence, by Weyl's inequality (Thm.B.3), we have:

$$\max\left\{\lambda_1(\tilde{\alpha}(\mathbf{X}\mathbf{X}^\top) \odot (\mathbf{X}\mathbf{X}^\top)), \lambda_1(\tilde{\beta}d\mathbf{X}\mathbf{X}^\top)\right\} \leq \lambda_1(\tilde{\mathbf{K}}) - \tilde{\gamma}d^2 \leq \lambda_1(\tilde{\alpha}(\mathbf{X}\mathbf{X}^\top) \odot (\mathbf{X}\mathbf{X}^\top)) + \lambda_1(\tilde{\beta}d\mathbf{X}\mathbf{X}^\top). \quad (16)$$

For $\mathbf{XX}^\top$, since Wishart matrix $\mathbf{W} = \frac{1}{d}\mathbf{XX}^\top$ follows Marchenko Pastur distribution (Marčenko & Pastur, 1967), it follows that

$$\lambda_1(\mathbf{XX}^\top) \to d(1 + \sqrt{n/d})^2 = n + d + 2\sqrt{nd}.$$

For $\mathbf{XX}^\top \odot (\mathbf{XX}^\top)$, we use our row-sum bound. The sum of $i$-th row of $(\mathbf{XX}^\top) \odot (\mathbf{XX}^\top)$ is:

$$\sum_{j=1}^{n}(\mathbf{x}_i^\top \mathbf{x}_j)^2 = \|\mathbf{x}_i\|^4 + \mathbf{x}_i^\top \left(\sum_{j \neq i} \mathbf{x}_j \mathbf{x}_j^\top\right) \mathbf{x}_i = \|\mathbf{x}_i\|^4 + \mathbf{x}_i^\top \mathbf{S}\mathbf{x}_i.$$

Here $\mathbf{S} \in \mathbb{R}^{d \times d}$ is independent of $\mathbf{x}_i$. The row sum follows the same distribution, hence all concentrating around:

$$\mathbb{E}_{\mathbf{x},\mathbf{S}}\left[\|\mathbf{x}\|^4 + \mathbf{x}^\top \mathbf{S}\mathbf{x}\right] = d^2 + 2d + \mathbb{E}_\mathbf{S}\mathrm{Tr}(\mathbf{S}) = dn + d^2 + d.$$

Therefore,

$$\lambda_1((\mathbf{XX}^\top) \odot (\mathbf{XX}^\top)) \approx dn + d^2 + d.$$

Substituting back to (16), we obtain that:

$$\tilde{\alpha}dn + (\tilde{\alpha} + \tilde{\gamma})d^2 + \tilde{\alpha}d \lesssim \lambda_1(\tilde{\mathbf{K}}) \lesssim (\tilde{\alpha} + \tilde{\beta})dn + (\tilde{\alpha} + \tilde{\beta} + \tilde{\gamma})d^2.$$

Equivalently,

$$\left(\frac{m}{4d} + \frac{1}{2}\right)n + \frac{m}{2} - \frac{m}{2\pi} + d - \frac{d}{2\pi} \lesssim \lambda_1(\tilde{\mathbf{K}}) \lesssim \left(\frac{m}{4d} + \frac{m}{2\pi d} + \frac{1}{2} + \frac{1}{2\pi}\right)n + \frac{m}{2} + d.$$

$\square$

## B.3. Estimating the Smallest Eigenvalue

In this section, we proceed to estimate the smallest NTK eigenvalue of the two-layer ReLU and ReGLU models. We begin by the canonical Schur product theorem (Horn & Johnson, 2012, Theorem 7.5.3).

**Theorem B.7** (Schur product theorem). *If both matrices $\mathbf{A}, \mathbf{B} \in \mathbb{R}^{n \times m}$ are positive semidefinite, then $\mathbf{A} \odot \mathbf{B}$ is also positive semidefinite.*

Furthermore, according to Pandit et al. (2024, Lemma 32), the limiting spectral distribution of $(\mathbf{XX}^\top) \odot (\mathbf{XX}^\top)$ is as follows.

**Lemma B.8** (Simplified). *Assume that $X_{ij} \overset{i.i.d}{\sim} \mathcal{N}(0,1)$, $n, d \to \infty$ and $\frac{2n}{d^2} \to \lambda$. Then the limiting spectral distribution of $\frac{1}{d^2}(\mathbf{XX}^\top) \odot (\mathbf{XX}^\top)$ follows Marchenko-Pastur distribution,*

$$\mu_\lambda = \begin{cases} (1 - \lambda^{-1})\delta_0 + \nu_\lambda, & \lambda > 1, \\ \nu_\lambda, & 0 < \lambda \leq 1. \end{cases}$$

*Here $\nu_\lambda$ is Marchenko-Pastur distribution with shape parameter $\lambda$.*

We now give our estimation of the smallest NTK eigenvalue.

**Proposition B.9** (Smallest eigenvalue). *Consider the two-layer ReLU and ReGLU models described in Sec.3.1.*

*1) For ReLU model, the smallest eigenvalue of its NTK matrix is given by*

$$\lambda_n(\mathbf{K}) \approx \frac{m+d}{4}(s^2 + 1) - \frac{m}{2\pi},$$

*where $s := \max\left\{0, 1 - \sqrt{n/d}\right\}$.*

*2) For ReGLU model, the smallest eigenvalue of its NTK matrix is given by*

$$\lambda_n(\tilde{\mathbf{K}}) \approx \frac{m+2d}{4}(\tilde{s}^2 + 1) + \frac{m+d}{2\pi}(s^2 - 1),$$

*where $\tilde{s} = \max\left\{0, 1 - \sqrt{2n}/d\right\}$.*

*Therefore, both $\lambda_n(\mathbf{K})$ and $\lambda_n(\tilde{\mathbf{K}})$ is of order $\Theta(m+d)$. When $n > d$, we have that*

$$\lambda_n(\tilde{\mathbf{K}}) > \lambda_n(\mathbf{K}).$$

*Proof.* **1) ReLU model.**

For ReLU mdoel, we recall that its NTK matrix can be written as

$$\mathbf{K} = \alpha \mathbf{X}\mathbf{X}^\top + \beta \mathbf{r}\mathbf{r}^\top + \gamma \mathbf{D}.$$

First, rank-1 update $\mathbf{r}\mathbf{r}^\top$ has negligible effect on the smallest eigenvalue of Wishart matrix. According to Horn & Johnson (2012, Corollary 4.3.9), we have

$$\lambda_n(\alpha \mathbf{X}\mathbf{X}^\top) \leq \lambda_n(\alpha \mathbf{X}\mathbf{X}^\top + \beta \mathbf{r}\mathbf{r}^\top) \leq \lambda_{n-1}(\alpha \mathbf{X}\mathbf{X}^\top).$$

When $n > d + 1$, by the property of Wishart matrix, $\lambda_n(\alpha \mathbf{X}\mathbf{X}^\top) = \lambda_{n-1}(\alpha \mathbf{X}\mathbf{X}^\top) = 0$, hence $\lambda_n(\alpha \mathbf{X}\mathbf{X}^\top + \beta \mathbf{r}\mathbf{r}^\top) = \lambda_n(\alpha \mathbf{X}\mathbf{X}^\top) = 0$.

When $n = d + 1$, we know that $\lambda_n(\alpha \mathbf{X}\mathbf{X}^\top) = 0, \lambda_{n-1}(\alpha \mathbf{X}\mathbf{X}^\top) = \alpha d \left(1 - \sqrt{\frac{d}{n}}\right)^2 \approx \frac{\alpha}{4n}$, hence $\lambda_n(\alpha \mathbf{X}\mathbf{X}^\top + \beta \mathbf{r}\mathbf{r}^\top) \lesssim \frac{\alpha}{4n}$ which is also negligible.

When $n < d + 1$, we consider the Wishart matrix. We define the classical location of Marchenko-Pastur distribution, $\gamma_i$, as follows,

$$1 - \frac{i}{n} = \int_0^{\gamma_i} \rho_{\mathrm{MP}}(x)\, dx,$$

where $\rho_{\mathrm{MP}}(x) = \frac{\sqrt{(x-a)(b-x)}}{2\pi c x}$ is the Marchenko-Pastur distribution. Here $c = n/d$ and the eigenvalues are supported on $[a, b]$, where $a = (1 - \sqrt{c})^2, b = (1 + \sqrt{c})^2$. Since we care about eigenvalues close to $a$, we may approximate the distribution as $\rho_{\mathrm{MP}} \approx C_a\sqrt{x - a}$, where $C_a = (\pi c^{3/4} a)^{-1}$. It follows from calculation that:

$$\gamma_{n-1} - \gamma_n \approx \left(\frac{3}{2 C_a n}\right)^{\frac{2}{3}} \left(2^{\frac{2}{3}} - 1\right) = O(n^{-\frac{2}{3}}).$$

Kafetzopoulos & Maltsev (2023, Theorem 3) give a rigidity bound concerning the eigenvalues of Wishart matrix, which states that $|\lambda_n - \gamma_n|, |\lambda_{n-1} - \gamma_{n-1}| \sim O\left(\frac{\log n}{n^2}\right)$. Therefore,

$$\lambda_{n-1} - \lambda_n \leq \gamma_{n-1} - \gamma_n + |\lambda_n - \gamma_n| + |\lambda_{n-1} - \gamma_{n-1}| \lesssim O(n^{\frac{2}{3}}).$$

This is also negligible.

Second, consider matrix $\alpha \mathbf{X}\mathbf{X}^\top$. Define $s = \max\left\{0, 1 - \sqrt{n/d}\right\}$, by property of Wishart matrix, we have that

$$\lambda_n(\alpha \mathbf{X}\mathbf{X}^\top) \to \alpha d s^2.$$

Finally, since the diagonal elements of $\mathbf{D}$ are sampled from the same distribution, we have that $D_{ii} \approx d$. Therefore,

$$\lambda_n(\mathbf{K}) \approx \frac{m+d}{4}(s^2 + 1) - \frac{m}{2\pi}.$$

**2) ReGLU model.**

For ReGLU model, recall that

$$\tilde{\mathbf{K}} = \tilde{\alpha}(\mathbf{X}\mathbf{X}^\top) \odot (\mathbf{X}\mathbf{X}^\top) + \tilde{\beta}(\mathbf{r}\mathbf{r}^\top) \odot (\mathbf{X}\mathbf{X}^\top) + \tilde{\gamma}\mathbf{D}^2. \tag{17}$$

First, we notice that $\mathbf{X}\mathbf{X}^\top$ is positive semidefinite. To see this, consider the Rayleigh quotient:

$$\lambda_n(\mathbf{X}\mathbf{X}^\top) = \min_{\|\mathbf{v}\|=1} \mathbf{v}^\top\mathbf{X}\mathbf{X}^\top\mathbf{v} = \|\mathbf{X}^\top\mathbf{v}\|^2 \geq 0.$$

Similarly, $\mathbf{r}\mathbf{r}^\top$ is also positive semidefinite. By Thm.B.7, $\mathbf{r}\mathbf{r}^\top \odot \mathbf{X}\mathbf{X}^\top$ is also positive semidefinite. Hence $\lambda_n(\mathbf{r}\mathbf{r}^\top \odot \mathbf{X}\mathbf{X}^\top) \geq 0$. By Weyl's inequality, we then have:

$$\lambda_n(\tilde{\alpha}(\mathbf{X}\mathbf{X}^\top) \odot (\mathbf{X}\mathbf{X}^\top) + \tilde{\beta}(\mathbf{r}\mathbf{r}^\top) \odot (\mathbf{X}\mathbf{X}^\top))$$
$$\geq \lambda_n(\tilde{\alpha}(\mathbf{X}\mathbf{X}^\top) \odot (\mathbf{X}\mathbf{X}^\top)) + \lambda_n(\tilde{\beta}(\mathbf{r}\mathbf{r}^\top) \odot (\mathbf{X}\mathbf{X}^\top))$$
$$\approx \lambda_n(\tilde{\alpha}(\mathbf{X}\mathbf{X}^\top) \odot (\mathbf{X}\mathbf{X}^\top)) + \lambda_n(\tilde{\beta}d(\mathbf{X}\mathbf{X}^\top)).$$

Second, we use the result proved by Pandit et al. (2024) (Lem.B.8). Define $\tilde{s} = \max\left\{0, 1 - \sqrt{2n}/d\right\}$, we have

$$\lambda_n\left(\tilde{\alpha}(\mathbf{X}\mathbf{X}^\top) \odot (\mathbf{X}\mathbf{X}^\top)\right) \to \tilde{\alpha}d^2\tilde{s}^2.$$

Finally, since the diagonal elements of $\mathbf{D}$ are sampled from the same distribution, we have that $D_{ii} \approx d$. Therefore,

$$\lambda_n(\tilde{\mathbf{K}}) \approx \frac{m+2d}{4}(\tilde{s}^2+1) + \frac{m+d}{2\pi}(s^2-1).$$

### 3) Comparing two eigenvalues.

Finally, we compare the two smallest eigenvalues. Subtracting one from another, we have

$$\lambda_n(\tilde{\mathbf{K}}) - \lambda_n(\mathbf{K}) \approx \frac{m+2d}{4}\tilde{s}^2 + \left(\frac{1}{4} - \frac{1}{2\pi}\right)[d - (m+d)s^2].$$

Because $n > d$, $s = 0$. Therefore,

$$\lambda_n(\tilde{\mathbf{K}}) - \lambda_n(\mathbf{K}) \approx \frac{m+2d}{4}\tilde{s}^2 + \left(\frac{1}{4} - \frac{1}{2\pi}\right)d > 0.$$

$\square$

## C. Proof of Loss Crossing Results

### C.1. Proof of Prop.4.1

Before proving Prop.4.1, we first prove the following lemma.

**Lemma C.1.** *For random vector $\mathbf{v} \in \mathbb{R}^n$ with mean $\boldsymbol{\mu}$ and coraviance matrix $\boldsymbol{\Sigma}$, for any fixed matrix $\mathbf{A} \in \mathbb{R}^{n\times n}$:*

$$\mathbb{E}_{\mathbf{v}}[\mathbf{v}^\top\mathbf{A}\mathbf{v}] = \mathrm{Tr}[\mathbf{A}\boldsymbol{\Sigma}] + \boldsymbol{\mu}^\top\mathbf{A}\boldsymbol{\mu}.$$

*Proof.* Using the properties of trace, we have

$$\mathbb{E}_{\mathbf{v}}[\mathbf{v}^\top\mathbf{A}\mathbf{v}] = \mathbb{E}_{\mathbf{v}}[\mathrm{Tr}(\mathbf{v}^\top\mathbf{A}\mathbf{v})] = \mathbb{E}_{\mathbf{v}}[\mathrm{Tr}(\mathbf{A}\mathbf{v}\mathbf{v}^\top)].$$

Since the trace is interchangeable with the expectation, we have that

$$\mathbb{E}_{\mathbf{v}}[\mathbf{v}^\top\mathbf{A}\mathbf{v}] = \mathrm{Tr}[\mathbb{E}_{\mathbf{v}}[\mathbf{A}\mathbf{v}\mathbf{v}^\top]]$$
$$= \mathrm{Tr}[\mathbf{A}\mathbb{E}_{\mathbf{v}}[(\mathbf{v}-\boldsymbol{\mu})(\mathbf{v}-\boldsymbol{\mu})^\top + \mathbf{v}\boldsymbol{\mu}^\top + \boldsymbol{\mu}\mathbf{v}^\top - \boldsymbol{\mu}\boldsymbol{\mu}^\top]]$$
$$= \mathrm{Tr}(\mathbf{A}\boldsymbol{\Sigma}) + \mathrm{Tr}(\mathbf{A}\boldsymbol{\mu}\boldsymbol{\mu}^\top)$$
$$= \mathrm{Tr}(\mathbf{A}\boldsymbol{\Sigma}) + \boldsymbol{\mu}^\top\mathbf{A}\boldsymbol{\mu}.$$

$\square$

**Proposition C.2.** *Consider the case when $m \to \infty$, i.e., kernel regime. Denote by $\mathbf{K}$ the NTK matrix, then the expected loss of the two-layer MLP evolves as*

$$\mathbb{E}_{\boldsymbol{\theta}}[L_k] \propto \operatorname{Tr}[(\mathbf{I} - \eta\mathbf{K})^{2k}\mathbf{K}] + \mathbf{Y}^{\top}(\mathbf{I} - \eta\mathbf{K})^{2k}\mathbf{Y}.$$

*Proof.* When $m \to \infty$, model operates in kernel regime. Hence, gradient descent gives:

$$\boldsymbol{\theta}_{k+1} = \boldsymbol{\theta}_k - \eta\nabla_{\boldsymbol{\theta}}L.$$

Consider MSE loss. In kernel regime, by (Lee et al., 2019), we have that

$$\mathbf{e}_{k+1} = (\mathbf{I} - \eta\mathbf{K})\mathbf{e}_k.$$

Here $\mathbf{e}_k = z_k(\mathbf{X}) - \mathbf{Y}$ is the residual at step $k$. Substituting this result to MSE loss, we have

$$
\begin{aligned}
L_k =& \frac{1}{2n}\|\mathbf{e}_k\|^2 \\
=& \frac{1}{2n}\mathbf{e}_{k-1}^{\top}(\mathbf{I} - \eta\mathbf{K})^2\mathbf{e}_{k-1} \\
=& ... \\
=& \frac{1}{2n}\mathbf{e}_0^{\top}(\mathbf{I} - \eta\mathbf{K})^{2k}\mathbf{e}_0.
\end{aligned}
$$

We know that infinite width limit yields gaussian process named NNGP (Lee et al., 2018). We have that:

$$\boldsymbol{\Sigma} = \mathbb{E}_{\boldsymbol{\theta}}[z_0(\mathbf{X})z_0(\mathbf{X})^{\top}].$$

Our analysis applies to both GLU and non-GLU structures. Here we consider two-layer non-GLU structure. In our two-layer model setting, we have that:

$$
\begin{aligned}
\Sigma_{ij} =& \mathbb{E}_{\boldsymbol{\theta}}[z_0(\mathbf{x}_i)z_0(\mathbf{x}_j)] \\
=& \sum_{k=1}^{m} \mathbb{E}_{\boldsymbol{\theta}}[\mathbf{V}_k^2\phi(\mathbf{W}_k^{\top}\mathbf{x}_i)\phi(\mathbf{W}_k^{\top}\mathbf{x}_j)] \\
=& m\sigma_v^2\mathbb{E}_{\mathbf{w}}[\phi(\mathbf{w}^{\top}\mathbf{x}_i)\phi(\mathbf{w}^{\top}\mathbf{x}_j)].
\end{aligned}
$$

For non-GLU NTK matrix expression (12), when $m \to \infty$, under LeCun initialization, the first term becomes dominant:

$$K_{ij} \approx m\mathbb{E}_{\mathbf{w}}[\phi(\mathbf{w}^{\top}\mathbf{x}_i)\phi(\mathbf{w}^{\top}\mathbf{x}_j)] = \Sigma_{ij}/\sigma_v^2.$$

Note that this result echoes "lazy training", where most training occurs at the output layer.

Now we can apply Lem.C.1. Since $z_0(\mathbf{X}) \sim \mathcal{N}(\mathbf{0}, \boldsymbol{\Sigma})$, $\mathbf{e}_0 \sim \mathcal{N}(-\mathbf{Y}, \boldsymbol{\Sigma})$. Hence,

$$\mathbb{E}_{\boldsymbol{\theta}}[L_k] \propto \operatorname{Tr}[(\mathbf{I} - \eta\mathbf{K})^{2k}\mathbf{K}] + \mathbf{Y}^{\top}(\mathbf{I} - \eta\mathbf{K})^{2k}\mathbf{Y}.$$

Moreover, define loss update $\Delta L_k := L_{k+1} - L_k < 0$, we have that for small learning rate $\eta$:

$$\mathbb{E}_{\boldsymbol{\theta}}[-\Delta L_k] \propto \eta[\operatorname{Tr}[(\mathbf{I} - \eta\mathbf{K})^{2k}\mathbf{K}^2] + \mathbf{Y}^{\top}(\mathbf{I} - \eta\mathbf{K})^{2k}\mathbf{K}\mathbf{Y}] + O(\eta^2).$$

Omitting $O(\eta^2)$ term, we have

$$\mathbb{E}_{\boldsymbol{\theta}}[-\Delta L_k] \propto \operatorname{Tr}[(\mathbf{I} - \eta\mathbf{K})^{2k}\mathbf{K}^2] + \mathbf{Y}^{\top}(\mathbf{I} - \eta\mathbf{K})^{2k}\mathbf{K}\mathbf{Y}.$$

$\square$

### C.2. Proof of Cor.4.2

As a corollary of Prop.4.1, we can provide a finer-grained analysis of training dynamics of the two-layer ReLU/ReGLU MLP models.

**Corollary C.3.** *Consider two-layer ReLU/ReGLU MLP models with gaussian inputs $\mathbf{x}_i \sim \mathcal{N}(\mathbf{0}, \mathbf{I}) \in \mathbb{R}^d$, we have that:*

*1) At early stage when $(\eta k)$ is small, as long as $\mathbf{Y}^\top(\mathbf{K} - \tilde{\mathbf{K}})\mathbf{Y} \geq 0, d \geq 5$ and $n \geq 300$, it holds that $\mathbb{E}_{\boldsymbol{\theta}}[L_k - \tilde{L}_k] < 0$. That is, ReLU model converges faster than ReGLU model.*

*2) At later stage when the minimum eigenvalue $\lambda_{\min}$ dominates the training process, for sufficiently large $k$, it holds that $\mathbb{E}_{\boldsymbol{\theta}}[L_k - \tilde{L}_k] > 0$. That is, ReGLU model takes over and converges faster than ReGLU model.*

*Proof.* **1) Early stage.**

We first consider the early stage when $(\eta k)$ is relatively small. Expanding the expression in Prop.4.1, we obtain

$$\mathbb{E}_{\boldsymbol{\theta}}[L_k] \propto \text{Tr}[(\mathbf{I} - \eta\mathbf{K})^{2k}\mathbf{K}] + \mathbf{Y}^\top(\mathbf{I} - \eta\mathbf{K})^{2k}\mathbf{Y}$$
$$\propto \text{Tr}(\mathbf{K}) + \mathbf{Y}^\top\mathbf{Y} - 2\eta k[\text{Tr}(\mathbf{K}^2) + \mathbf{Y}^\top\mathbf{K}\mathbf{Y}] + O(\eta^2 k^2).$$

Similarly, for two-layer ReGLU model, we have

$$\mathbb{E}_{\boldsymbol{\theta}}[\tilde{L}_k] \propto \text{Tr}[(\mathbf{I} - \eta\tilde{\mathbf{K}})^{2k}\tilde{\mathbf{K}}] + \mathbf{Y}^\top(\mathbf{I} - \eta\tilde{\mathbf{K}})^{2k}\mathbf{Y}$$
$$\propto \text{Tr}(\tilde{\mathbf{K}}) + \mathbf{Y}^\top\mathbf{Y} - 2\eta k[\text{Tr}(\tilde{\mathbf{K}}^2) + \mathbf{Y}^\top\tilde{\mathbf{K}}\mathbf{Y}] + O(\eta^2 k^2).$$

Therefore, omitting high order terms, we have

$$\mathbb{E}_{\boldsymbol{\theta}}[L_k - \tilde{L}_k] \propto \text{Tr}(\mathbf{K} - \tilde{\mathbf{K}}) - 2\eta k[\text{Tr}(\mathbf{K}^2 - \tilde{\mathbf{K}}^2) + \mathbf{Y}^\top(\mathbf{K} - \tilde{\mathbf{K}})\mathbf{Y}]. \tag{18}$$

There are three terms in above expression. By assumption, we have that $\mathbf{Y}^\top(\mathbf{K} - \tilde{\mathbf{K}})\mathbf{Y} \geq 0$. Consider $\text{Tr}(\mathbf{K} - \tilde{\mathbf{K}})$. According to Eq.(14) and Eq.(15), at the infinite width limit we have that

$$K_{ii} = \frac{m}{2d}\|\mathbf{x}_i\|^2, \qquad \tilde{K}_{ii} = \frac{m}{2d^2}\|\mathbf{x}_i\|^4.$$

Therefore, according to law of large numbers,

$$\begin{aligned}
\text{Tr}(\mathbf{K} - \tilde{\mathbf{K}}) &= \sum_{i=1}^n (K_{ii} - \tilde{K}_{ii}) \\
&= \frac{nm}{2d}\mathbb{E}_{\mathbf{x}}\left[\|\mathbf{x}\|^2 - \frac{\|\mathbf{x}\|^4}{d}\right] \\
&= \frac{nm}{2d}\left(d - \frac{d(d+2)}{d}\right) \\
&= -\frac{nm}{d} < 0.
\end{aligned}$$

Here we use the fact that, for standard gaussian vector $\mathbf{x} \sim \mathcal{N}(\mathbf{0}, \mathbf{I})$,

$$\mathbb{E}_{\mathbf{x}}[\|\mathbf{x}\|^k] = 2^{k/2}\frac{\Gamma\left(\frac{d+k}{2}\right)}{\Gamma\left(\frac{d}{2}\right)},$$

where $\Gamma$ is gamma function.

Finally, consider the second term, we have that

$$\text{Tr}(\mathbf{K}^2 - \tilde{\mathbf{K}}^2) = \sum_{i=1}^n \sum_{j=1}^n K_{ij}^2\left(1 - \frac{(\mathbf{x}_i^\top\mathbf{x}_j)^2}{d^2}\right).$$

When $i = j$, we have that

$$\sum_{i=1}^{n} K_{ii}^2 \left(1 - \frac{\|\mathbf{x}_i\|^4}{d^2}\right)$$

$$= \frac{nm^2}{4d^2} \mathbb{E}_{\mathbf{x}} \left[\|\mathbf{x}\|^4 - \frac{\|\mathbf{x}\|^8}{d^2}\right]$$

$$= -\frac{nm^2}{2d^3}(d+2)(5d+12).$$

When $i \neq j$,

$$\sum_{1 \le i \neq j \le n} K_{ij}^2 \left(1 - \frac{(\mathbf{x}_i^\top \mathbf{x}_j)^2}{d^2}\right)$$

$$= \frac{n(n-1)m^2}{4d^2} \mathbb{E}_{\mathbf{x},\mathbf{x}'} \left[\left(\frac{(\mathbf{x}^\top \mathbf{x}')}{2} + \frac{\|\mathbf{x}\|\|\mathbf{x}'\|}{\pi}\right)^2 \left(1 - \frac{(\mathbf{x}^\top \mathbf{x}')^2}{d^2}\right)\right]$$

$$= \frac{n(n-1)m^2}{4d^3} \left(\frac{d^2 - 3d - 6}{4d} + \frac{d^3 - d^2 - 4d - 4}{\pi^2 d}\right).$$

Putting together, we have:

$$\text{Tr}(\mathbf{K}^2 - \tilde{\mathbf{K}}^2) = \frac{nm^2}{4d^3} \left[(n-1)\left(\frac{d^2 - 3d - 6}{4d} + \frac{d^3 - d^2 - 4d - 4}{\pi^2 d}\right) - 10d^2 - 44d - 48\right].$$

One can verify that, when $d \ge 5$ and $n \ge 300$, $\text{Tr}(\mathbf{K}^2 - \tilde{\mathbf{K}}^2) > 0$.

Since $\text{Tr}(\mathbf{K} - \tilde{\mathbf{K}}) < 0$, $\text{Tr}(\mathbf{K}^2 - \tilde{\mathbf{K}}^2) > 0$ and $\mathbf{Y}^\top(\mathbf{K} - \tilde{\mathbf{K}})\mathbf{Y} \ge 0$, according to (18), $\mathbb{E}_{\boldsymbol{\theta}}[L_k - \tilde{L}_k] < 0$. In addition, when $k$ increases, $\mathbb{E}_{\boldsymbol{\theta}}[L_k - \tilde{L}_k]$ would further decrease.

**2) Later stage.**

To further analyze the dynamics at later stage, we use eigendecomposition. Denote by $(\lambda_i, \mathbf{v}_i)$ the eigenpairs of $\mathbf{K}$ and $\beta_i := \mathbf{Y}^\top \mathbf{v}_i$, for ReLU model, we have that:

$$\mathbb{E}_{\boldsymbol{\theta}}[L_k] \propto \text{Tr}[(\mathbf{I} - \eta\mathbf{K})^{2k}\mathbf{K}] + \mathbf{Y}^\top(\mathbf{I} - \eta\mathbf{K})^{2k}\mathbf{Y}$$

$$\propto \sum_{i=1}^{n}(\lambda_i + \beta_i^2)(1 - \eta\lambda_i)^{2k}.$$

At later stage when $k \to \infty$, $(1 - \eta\lambda_n)^{2k}$ demonates the above expression. We have,

$$\mathbb{E}_{\boldsymbol{\theta}}[L_k] \propto (\lambda_n + \beta_n^2)(1 - \eta\lambda_n)^{2k} \sum_{i=1}^{n} \frac{\lambda_i + \beta_i^2}{\lambda_n + \beta_n^2} \left(\frac{1 - \eta\lambda_i}{1 - \eta\lambda_n}\right)^{2k}$$

$$\sim (\lambda_n + \beta_n^2)(1 - \eta\lambda_n)^{2k}$$

Similarly, for ReGLU model

$$\mathbb{E}_{\boldsymbol{\theta}}[\tilde{L}_k] \sim (\tilde{\lambda}_n + \tilde{\beta}_n^2)(1 - \eta\tilde{\lambda}_n)^{2k}.$$

Now, solving for $(\lambda_n + \beta_n^2)(1 - \eta\lambda_n)^{2k} > (\tilde{\lambda}_n + \tilde{\beta}_n^2)(1 - \eta\tilde{\lambda}_n)^{2k}$, we obtain that

$$k > \frac{\log[(\tilde{\lambda}_n + \tilde{\beta}_n^2)/(\lambda_n + \beta_n^2)]}{2\log[(1 - \eta\lambda_n)/(1 - \eta\tilde{\lambda}_n)]}.$$

Hence, for sufficiently large $k$, it holds that $\mathbb{E}_{\boldsymbol{\theta}}[L_k - \tilde{L}_k] > 0$.

$\square$

# D. Experiment on Generalization Gap Analysis

In this section, we provide more experiment details and results on our generalization gap analysis.

**Models.** We use MLP Mixer (Tolstikhin et al., 2021), ViT (Dosovitskiy et al., 2021) and GPT-2 (Radford et al., 2019) in our experiments. MLP Mixer and ViT are used for image classification tasks, and GPT-2 is used for natural language processing (NLP) task. All three models use MLP in their architecture design, allowing us to compare their ability in reducing generalization gap with and without GLU structure.

**Datasets.** For image classification task, we use Tiny ImageNet (Le & Yang, 2015) and CIFAR-10 (Krizhevsky, 2009). Standard data augmentation are applied, including random resized cropping, color jittering, horizontal flipping, and random erasing for Tiny ImageNet dataset, and random cropping with padding, horizontal flipping, and random erasing for CIFAR-10 dataset. For NLP task, we use FineWeb-Edu (Penedo et al., 2024).

**Optimizer and regularization.** For MLP Mixer and ViT, we use traditional stochastic gradient descent (SGD) with no additional regularization. The learning rate is set to $5 \times 10^{-3}$ for MLP Mixer and $0.01$ for ViT. For GPT-2, we follow standard training method, using AdamW optimizer with a learning rate of $4 \times 10^{-4}$ and weight decay of $0.1$.

Below are plots showing the generalization gap $L_{\mathcal{D}} - L_S$ against the training loss $L_S$ for models with and without GLU. All experiments are performed on NVIDIA 3090 GPU with 24GB of memory. From these results we can observe that for the same training error, GLU structure has limited influence on reducing the generalization gap, solidifying our argument that the advantage of GLU structure mainly comes from accelerating optimization.

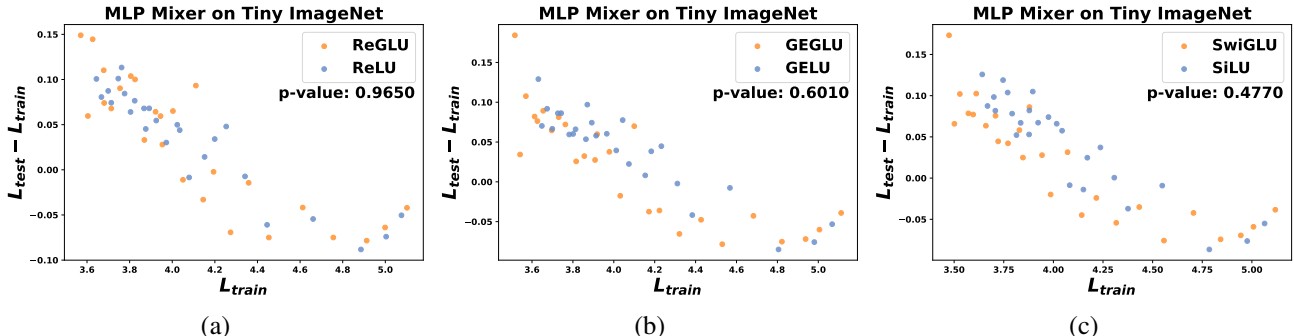

*Figure 8.* Generalization gap vs training loss with MLP Mixer trained on Tiny ImageNet. (a) ReLU vs ReGLU. (b) GELU vs GEGLU. (c) SiLU vs SwiGLU.

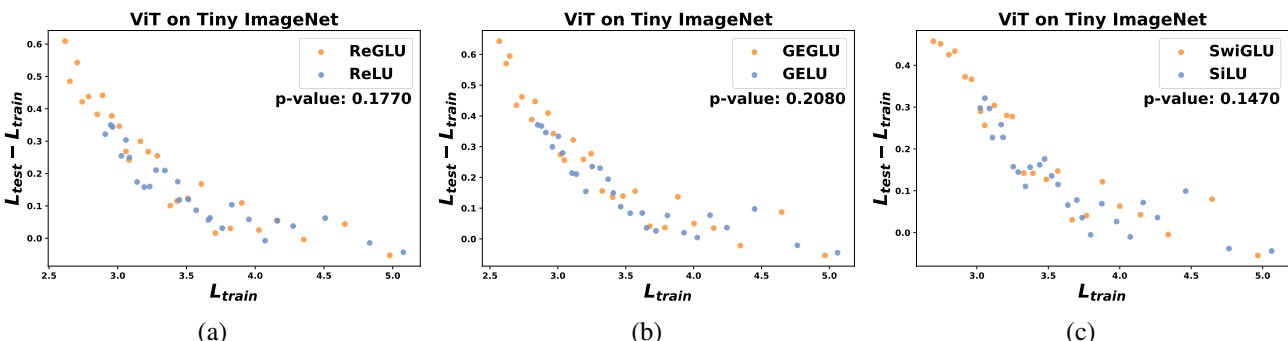

*Figure 9.* Generalization gap vs training loss with ViT trained on Tiny ImageNet. (a) ReLU vs ReGLU. (b) GELU vs GEGLU. (c) SiLU vs SwiGLU.

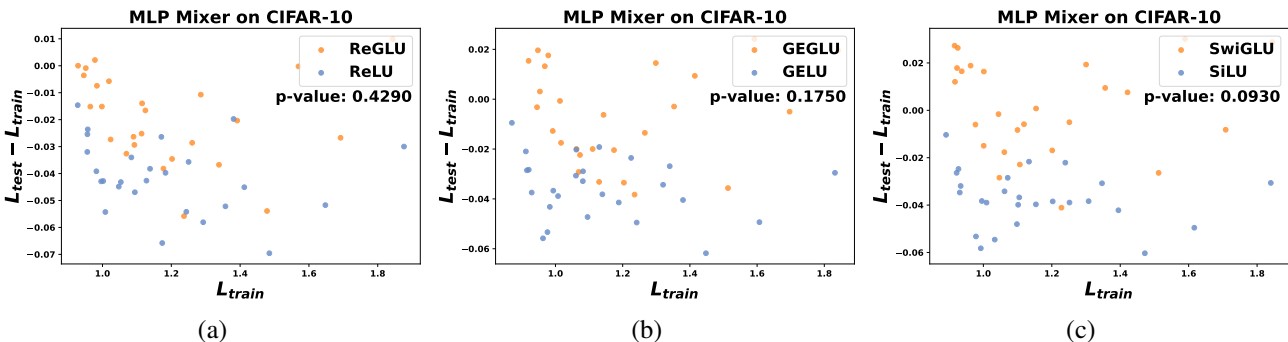

*Figure 10.* Generalization gap vs training loss with MLP Mixer trained on CIFAR-10. (a) ReLU vs ReGLU. (b) GELU vs GEGLU. (c) SiLU vs SwiGLU.

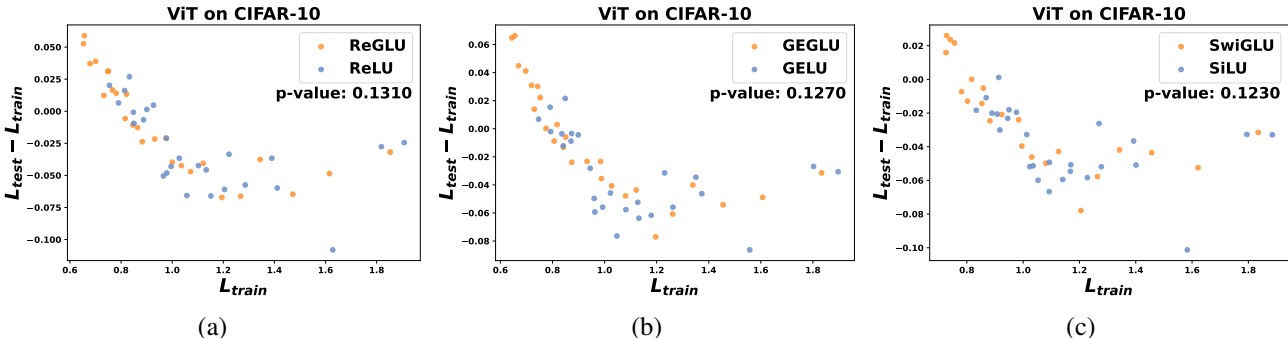

*Figure 11.* Generalization gap vs training loss with ViT trained on CIFAR-10. (a) ReLU vs ReGLU. (b) GELU vs GEGLU. (c) SiLU vs SwiGLU.

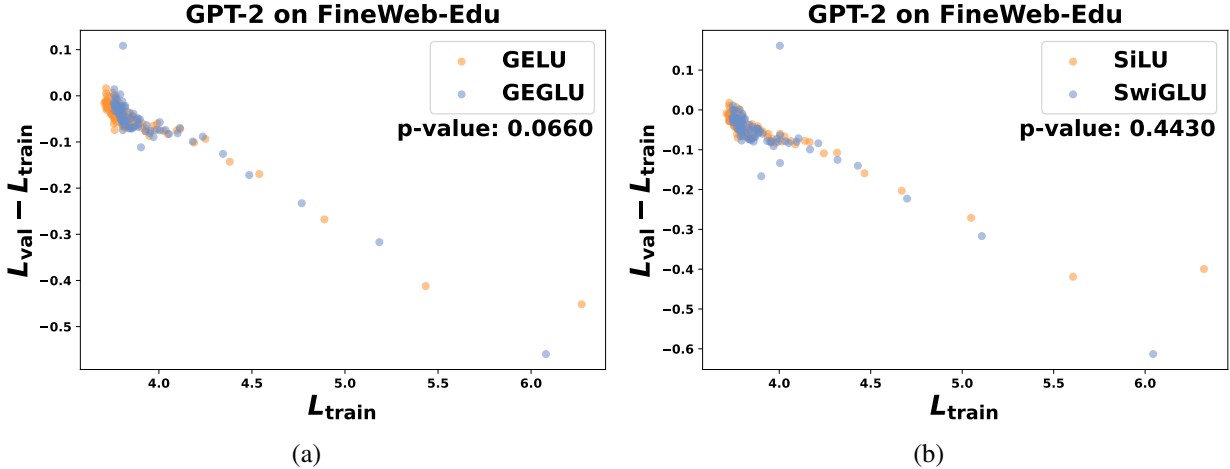

*Figure 12.* Generalization gap vs training loss with GPT-2 trained on FineWeb-Edu. (a) GELU vs GEGLU. (b) SiLU vs SwiGLU.

