# OpenReview forum: "The Devil is in the Condition Numbers: Why is GLU Better than non-GLU Structure?"
_ICML.cc/2026/Conference — ICML 2026 regular_

### Official Review · Reviewer_7bXQ · 2026-03-07

**Soundness:** 2
**Presentation:** 2
**Significance:** 3
**Originality:** 3
**Overall Recommendation:** 4
**Confidence:** 2

**Summary:**

This paper analyzed the effect of GLU on accuracy.
Specifically, the paper analyzed this by decomposing the generalization error into training errors and stochastic errors.
They proposed a novel approach to analyze the two using the NTK spectrum.
Through theoretical analysis, they find that the NTK spectrum of the GLU variant has a lower condition number and a more compact eigenvalue distribution.
With this perspective, they conclude that the GLU leads to better convergence in training loss.
They provide empirical results on ViT and GPT-2, suggesting that GLU's primary benefit lies in accelerating optimization rather than reducing stochastic error.

**Compliance With Llm Reviewing Policy:**

Affirmed.

**Final Justification:**

The rebuttal has resolved the uncertainty I had about the original manuscript, and the study's contribution is now clear to me.
So, I will raise my rating from weak-reject to weak-accept.

**Key Questions For Authors:**

## Discussion in  "accelerated convergence of training error."
The authors note that using GLU leads to "accelerated convergence of training error."
In Fig.1 and in the loss-crossing phenomenon analysis, the loss of non-GLU variants drops faster at the beginning of the training (c.f. before crossing).
In line 387, the author states that  "diminishing the early-stage advantage of non-GLU models".
Based on this discussion, I understand that the author discusses "accelerated convergence"  after the crossing.
If this is the case, does the author conclude that both reach the same training loss when one continues training? (because GLU only accelerated convergence, not the final fitting)
Are there any theoretical results from NTK analysis on the convergence (e.g., both reach a similar error)? Or are there any empirical results to support this?
In fig. 6, GLU variants achieve lower training loss at the very end of training, but I could not find any sign that both reach the same training loss.
(I suspect that I might misunderstand some important discussion or the definition of "convergence".)

## Impact of number of parameters for training loss.
When the number of input/output channels is the same, GLU variants have twice as many parameter counts as non-GLU variants.
Therefore, adopting the GLU not only introduces second-order nonlinearity (as the authors discuss) but also improves fitting capabilities by increasing the number of parameters.
How does the author choose the parameter count? (I want to know, if they adopted the same input/output channel count, how did the authors remove the influence of the additional parameter count in the empirical analysis?

**Limitations:**

yes

**Strengths And Weaknesses:**

# Strength
## Interesting and  analysis
Their analysis of the effect of GLU, which accelerates convergence in training error but has little effect on stochastic error, is very interesting.
This is, in some sense, counterintuitive and gives the researcher in this field a new insight into the standard module of GLU.

## Theoritical
A thorough mathematical analysis theoretically supports the proposed approach using NTK to analyze the convergence.

# Weakness
Although the finding and analysis looks very interesting.
Some part of this discussion, which relates to the main claim of the paper that the GLU "accelerates convergence in training error" and the important setting for the empirical analysis, may need more clarification.
Please refer to the "Key Questions For Authors" section for the specific point that I found difficult to understand.

---

> ### Author Rebuttal · Authors · 2026-03-31
>
> > **Q1:** I understand that the author discusses "accelerated convergence" after the crossing. If this is the case, does the author conclude that both reach the same training loss when one continues training?
>
> **A1:** Thank you for this important question. We agree that the wording should be made more precise. What we mean is **a faster overall optimization process under a finite training budget**, rather than saying that GLU is always faster either strictly before or strictly after the crossing point. The loss-crossing phenomenon is consistent with this interpretation: non-GLU may decrease faster at the very beginning, while GLU benefits later from the better-conditioned spectrum and achieves lower training loss within the same training budget.
>
> We also do not claim that GLU and non-GLU must converge to the same final training loss. **Since the two architectures are different, they generally induce different loss landscapes and may converge to different loss values**. In this sense, our point is that under the same practical budget GLU tends to reach a better-trained model.
>
> Empirically, our finite training budget experiments show that GLU variants achieve lower training loss by the end of training, which is exactly the practical setting we aim to explain. We will revise the wording in the paper to make this distinction explicit.
>
> > **Q2:** GLU introduces more parameters than non-GLU. How does the author choose the parameter count?
>
> **A2:** Thank you for raising this important concern. In fact, for fair comparison, we use **reduced intermediate dimensions in GLU-based models** so that total parameter counts are comparable to non-GLU baselines, which is also standard practice in transformer architectures [1][2][3]. Furthermore, training budgets are kept consistent across variants.
>
> We acknowledge that this was not clearly described and will revise the manuscript accordingly.
>
> [1] Shazeer, GLU Variants Improve Transformer. (2020)
>
> [2] Narang et al., Do Transformer Modifications Transfer Across Implementations and Applications? (2021)
>
> [3] Qiu et al., Gated Attention for Large Language Models: Non-linearity, Sparsity, and Attention-Sink-Free. (2025)

---

> > ### Author Rebuttal · Reviewer_7bXQ · 2026-04-01
> >
> > Thank you for the clarification. All my concerns have been resolved.

---

> > > ### Author Response · Authors · 2026-04-01
> > >
> > > Thank you very much for the careful follow-up! We are very glad that our clarification has resolved your concerns. Since all of your concerns have been addressed, we would be very grateful if you could consider increasing your score accordingly :)

---

### Official Review · Reviewer_vHko · 2026-03-11

**Soundness:** 3
**Presentation:** 2
**Significance:** 3
**Originality:** 3
**Overall Recommendation:** 3
**Confidence:** 3

**Summary:**

The paper explores why GLU-type activations perform better than non-gated alternatives; the primary findings relates to how these functions lead to matrices with smaller condition numbers within the NTK spectrum, leading to more stable matrices and faster convergence.

**Compliance With Llm Reviewing Policy:**

Affirmed.

**Final Justification:**

I appreciate the response from the authors; while I believe the direction is interesting, I still find a bit of a disconnect between some of the direct claims the authors make on possibly larger and more complex models given the current form of the analysis. As such, I am still more inclined to retain my score at the current confidence. Nevertheless, I am more than willing to support acceptance a consensus is reached among the other reviewers or if the area chair is more inclined towards doing so.

**Key Questions For Authors:**

- GLU and non-GLU variants often differ in the number of parameters, which can be significant in attention models. How do the authors control for this between their experiments, as this is a rather important detail not mentioned?

**Limitations:**

Yes these are discussed.

**Strengths And Weaknesses:**

Strengths:
- The paper is well written and despite the somewhat dense content in terms of mathematical background, the authors do a good job of explaining the most important details in a relatively concise manner.
- The general theoretical details are supported by some experimental results, which is nice to observe.

Weaknesses:
- For the experimental results, I am somewhat unsure if only experimenting on ViT/GPT-2 is enough to draw major conclusions, as there have been significantly architectural changes that have come throughout the years (pre/post-layernorm, positional embeddings, types of attention structures/patterns) that could elicit the need to repeat some of the same experiments to confirm the general presence of the observed effects and not just as a side-effect of potential non-ideal design choices, particularly given the age of the two models.
- Additionally, I do think some experiments could focus on details that could focus on things on which the change in the condition number could have a more outsized effect. For example, I do believe that concepts such as attention sinks or rank collapse would be much more relevant than the stochastic error analysis section (which is interesting but not particularly revealing and something I think has already been implicitly explored in other works; it would probably fit better as a much smaller ablation).
- My primary takeaway from this work is that the authors provide a hypothesis for why GLU activations can help stabilize or accelerate model training; while this is interesting, I think years of more empirical research in the community has already revealed this to be the case. Explanations and new perspectives are always interesting, but I do find the direct messaging slightly hard to reconcile since the analysis is for a 2 layer model and the experiments appear to be for much deeper ones; quite a lot of analysis has already been done in the past to show that deeper models behave much differently, particularly attention-based models. This, in my opinion, is a rather large issue that should be addressed more carefully.

---

> ### Author Rebuttal · Authors · 2026-03-31
>
> ## Figures and Tables: https://anonymous.4open.science/api/repo/attachment-7234/file/attachment.pdf
>
> > **Q1:** Insufficient experiment on architectural changes.
>
> **A1:** We added three types of evidence.
>
> First, we extended the model-family comparison to more recent LLMs, including Qwen, Llama, and DeepSeek; see our response to reviewer `Lk7y` (`A3`). Across these families, GLU variants again show **smaller NTK condition numbers together with better optimization**.
>
> Second, we directly tested architectural ablations over normalization strategy, positional encoding, and attention mechanism. The results are summarized in **Table 4A** at the attachment `link`, where GLU consistently achieves **both lower NTK condition numbers and lower final training loss than its non-GLU counterpart**. For example, under absolute positional encoding, the condition number still drops from 30.17 to 25.16 and the final training loss from 2.924 to 2.776.
>
> Third, to connect the theory with deeper models, we further analyzed the case with preceding feature module $u=g(x)$; see `A4` below. Taken together, these results suggest that the effect is not tied to specific design choice.
>
> > **Q2:** Attention sink or rank collapse are more relevant than stochastic error analysis.
>
> **A2:** For representation collapse, **Table 5A** at the attachment `link` shows a consistent pattern across all three model families: **GLU yields higher effective rank and lower top singular-value concentration**. For instance, on DeepSeekV3 the effective rank increases from 99.46 to 126.67, while the top singular-value mass decreases from 0.0744 to 0.0709. This suggests less severe collapse in the final hidden representations.
>
> For attention sinks, **Table 6A** at the attachment `link` reports attention mass on the first token, cumulative mass on the first four positions, attention entropy, and maximum attention mass. Here the effect is **not consistent** across families (e.g., sink-first4 decreases on Qwen and DeepSeek but not on Llama.), suggesting that GLU has no consistent influence on attention sink behavior.
>
> Finally, we also want to emphasize that **rank collapse and attention sinks are intermediate phenomena**, whereas our stochastic-error analysis addresses the paper's main outcome-level question: whether GLU mainly improves optimization or also materially changes the stochastic error.
>
> > **Q3:** Research has already revealed the reason behind GLU.
>
> **A3:** We agree that the empirical fact that GLU improves training and final performance has been widely observed. Our point is that they do not provide a mechanism-level explanation for **why FFN GLU improves optimization**. For example, [1] mainly document empirical gains and transferability across settings, while [2][3] study gated attention rather than the FFN GLU structure considered here. We note that the theoretical gap on GLU is recently also explicitly acknowledged in [4], which notes that the underlying principles behind GLU remain open.
>
> Therefore, our goal is not to rediscover that GLU helps training, but to explain this known phenomenon through NTK spectral reshaping and improved conditioning.
>
> [1] Shazeer, GLU Variants Improve Transformer (2020)
>
> [2] Wang, GLU Attention Improve Transformer (2025)
>
> [3] Qiu et al., Gated Attention for Large Language Models: Non-linearity, Sparsity, and Attention-Sink-Free (2025)
>
> [4] Zhong et al., Understanding Transformer from the Perspective of Associative Memory (2025)
>
> > **Q4:** Deeper models behave much differently.
>
> **A4:** We additionally analyzed the model with preceding feature module $u=g(\theta; x)$. Consider non-GLU model $z=V\phi(Wu)$, GLU model $z=V[(Pu)\odot \phi(Wu)]$ and corresponding NTK matrix $K, \tilde K$. After derivation, we have
> $$
> \tilde K \approx K\odot (UU^{\top}/d) + K_g \odot (\Sigma_{\phi}/d),
> \tag{*}
> $$
> where $K_g$ is the NTK of the module $g$, and $\Sigma_{\phi} = \mathbb{E}[\phi(w^{\top}x)\phi(w^{\top}x')]$. This expression suggests two points. First, when normalization keeps $\\lvert|u_i\\rvert| \approx \sqrt{d}$ , $UU^{\top}/d$ is close to cosine-similarity matrix, so **the first term preserves the same diagonal-dominance effect as in our two-layer analysis**. Second, the preceding layers enter through $K_g$, meaning that **GLU not only reweights the last-layer kernel but also inherits the improved geometry from earlier layers**.
>
> We also verified this mechanism empirically beyond the plain two-layer setting. For both an MLP-based and an attention-based feature module, the approximation in Eq. $(*)$ matches the empirical NTK quite well, see **Table 7A and 8A** at the attachment `link`. Also, GLU continues to reduce condition number in our architecture ablations (see `A1` above) and in the additional model-family experiments reported in **Table 2A**.
>
> > **Q5:** How do the authors control parameter count?
>
> **A5:** Please see our response `A2` to reviewer `7bXQ`.

---

> > ### Author Rebuttal · Reviewer_vHko · 2026-04-02
> >
> > I thank the authors for their response.
> >
> > While the analysis the authors provide is interesting, it ultimately doesn't come across as very useful; GLU-type activation have long been empirically studied as being more effective in large models such as Transformers. The hypothesis regarding the NTK spectrum is interesting but many of the experiments ultimately still fails to completely isolate the effect of the GLU activation itself; I believe doing so would really require a more mechanistic approach where activations are individually modified or so.
> >
> > Additionally, after re-reading some of the theory provided in more detail, it would appear that much of it still remains largely hard to validate given the assumptions in Section 3.1.
> >
> > For this reason, I am more inclined to maintain my score.

---

> > > ### Author Response · Authors · 2026-04-02
> > >
> > > Thank you for your follow-up. We would like to clarify why we believe the present findings remain useful and relevant.
> > >
> > > > **Q1:** GLU-type activation has long been empirically studied as being more effective in large models such as Transformers.
> > >
> > > **A1:** We agree that the empirical effectiveness of GLU has been widely observed. However, the fact that a phenomenon is empirically well known does not make a mechanism-level explanation unimportant; rather, it is precisely what makes such an explanation valuable. In fact, prior work has explicitly highlighted that the effectiveness of GLU remains insufficiently understood. For example, [1] famously described GLU's empirical success as a kind of "divine benevolence", and recently [2] closes their discussion of SwiGLU in Section 2.1.2 by stating: "As for why gating mechanisms are effective in general, we encourage further research and exploration into their underlying principles and theoretical explanations." Our work is aimed exactly at this open explanatory gap.
> > >
> > > Furthermore, beyond the Transformer results already discussed in the paper and attachment, we also observe the same trend on MLP-Mixer, which has no attention mechanism at all: **GLU reduces condition number and improves optimization there as well** (see the three figures in https://anonymous.4open.science/api/repo/attachment-7234/file/mixer_figures.pdf). Together with our architectural ablations (see **Table 4A** at attachment `link`), this suggests that the phenomenon is broad: across small and large models, across different Transformer designs, and even in non-attention architectures, **replacing the FFN by its GLU counterpart consistently improves conditioning and optimization**.
> > >
> > > [1] Shazeer, GLU Variants Improve Transformer (2020)
> > >
> > > [2] Zhong et al., Understanding Transformer from the Perspective of Associative Memory (2025)
> > >
> > > > **Q2:** Experiments still fail to isolate the effect of the GLU activation.
> > >
> > > **A2:** To make the test more mechanistic, we additionally run an interpolation experiment that continuously varies the strength of the GLU operation while keeping the surrounding architecture (attention, positional embedding and etc.) fixed. Concretely, we consider the interpolated case
> > > $$
> > > z_\alpha = V[((1-\alpha)\mathbf{1} + \alpha Px)\odot \phi(Wx)],
> > > \qquad \alpha\in[0,1].
> > > $$
> > > Thus $\alpha=0$ recovers the non-GLU structure, $\alpha=1$ recovers the full GLU structure. Our experiment results are shown below:
> > >
> > > | value of $\alpha$ | Final Train Loss | Condition Number |
> > > |:---|---:|---:|
> > > | 0.00 | 2.4756 | 2.2970 |
> > > | 0.25 | 2.4300 | 2.0055 |
> > > | 0.50 | 2.3743 | 1.6769 |
> > > | 0.75 | 2.3111 | 1.4575 |
> > > | 1.00 | 2.2770 | 1.3936 |
> > >
> > > As $\alpha$ increases from 0 to 1, **the training loss and the condition number both monotonically decrease**. This is exactly the kind of trend one would expect if the gain is genuinely tied to the GLU operation itself rather than to an unrelated side effect, and aligns with our theoretical conclusion.
> > >
> > > Beyond this direct interpolation test, our original architectural ablations already change normalization, positional encoding, and attention mechanism while keeping the FFN replacement itself fixed; across all of these settings, GLU still consistently lowers both condition number and training loss.
> > >
> > > > **Q3:** Much of the theory remains hard to validate given the assumptions in Section 3.1.
> > >
> > > **A3:** First, we would like to stress again that both the two-layer NTK relation
> > > $$
> > > \tilde K\approx K\odot (XX^{\top}/d)
> > > $$
> > > and our newly developed relation for deep models with an FFN layer
> > > $$
> > > \tilde K\approx K\odot (UU^{\top}/d) + K_g\odot (\Sigma_u/d)
> > > $$
> > > **do not require Gaussian i.i.d. inputs or ReLU/ReGLU activation assumptions**. These assumptions are introduced only later, at the theorem level, to obtain explicit quantitative spectral bounds. In other words,
> > >
> > > > **the assumptions in Section 3.1 (specifically, Theorem 3.1) are used to make the spectral properties quantitative, not to create the better conditioning mechanism itself.**
> > >
> > > Second, once either of the above two relations holds, the reweighting factor suppresses off-diagonal correlations and makes $\tilde K$ more diagonally dominant than $K$, which is precisely the mechanism behind improved conditioning. This point is directly supported by the empirical validations in **Table 1A, Table 7A, and Table 8A**, and by the fact that the lowered condition number and improved optimization continue to hold beyond the Section 3.1 assumptions; see **Figure 3 and Figure 6** in the main text, as well as **Table 2A, Table 3A, and Figure 1A** in the attachment `link`.
> > >
> > > For this reason, we believe it is not accurate to say that the theory is "largely hard to validate." The assumptions restrict the scope of the quantitative theorem, but the core kernel relations and their predicted optimization consequences have in fact been validated empirically.

---

### Official Review · Reviewer_kQfg · 2026-03-11

**Soundness:** 3
**Presentation:** 4
**Significance:** 4
**Originality:** 3
**Overall Recommendation:** 5
**Confidence:** 4

**Summary:**

The paper analyzed a two layer MLP and compared it against the GLU-variant.
They were able to very rigorously prove that the NTK, given some typical assumptions in these sort of analysis, is more well-conditioned by switching to GLU when looking at ReLU.
Under the same framework, they also explained the crossing over behavior that can be seen in loss plots.
Finally, they argued that GLU variants are strictly only training better, and not introducing any better inductive biases or generalization.

**Compliance With Llm Reviewing Policy:**

Affirmed.

**Final Justification:**

My original concerns were relatively minor for the paper, and the rebuttal has resolved them. I am maintaining my score of accept.

**Key Questions For Authors:**

1. How is "stochastic error" different from validation/test set metrics? I'm a bit confused as to how this was calculated empirically. See also my comment regarding the increased parameter count for GLU above.
2. Line (205) $d+1<n$, where is this used in the proofs?
3. Do these results mean that using Muon (or other optimizers which relies on the whitening metric) on GLU would be less useful since the geometry is already "more" structured? e.g. in the silly case where some hypothetical activation function makes NTK identity, then SGD works perfectly fine.
4. Is the generalization of the proofs to other activation functions was not done is strictly due to the availability of the arc-cosine kernel?
5. Equation (16) in appendix is a bit unclear; can you elaborate on how you got that from B.3? Similarly, can you elaborate on the structure of $S$ on lines 889.

**Limitations:**

Limitations should be discussed a bit more: e.g. NTK analysis done for 2 layer MLP only, when NTK assumptions break down, the infinite width case versus the actual implemented cases.

**Strengths And Weaknesses:**

Strengths:
- The main text is very well written and easy to follow with notation that overall doesn't obfuscate the results.
- Rigorous yet approachable proofs for their main results
- Arguably a significant forward progress on Shazeer's "divine benevolence" attribution.

Weaknesses:
- Experimental conditions not all fully described; hard to impossible to replicate given just the paper + appendices. For example, in Figure 2, how big are the dimensions? We see in the proof that the approximations are only valid as one takes some limits; showing where this breaks down might be interesting. Furthermore, plotting the actual eigenvalues and showing the largest individually and smallest individually obeys Tab. 1 scaling would be convincing. Just overall concerns regarding reproducibility in the examples.
- In section 5, I'm not sure if the runs are accounting for the additional parameters/FLOPs that GLU introduces. e.g. more parameters might need more run time?
- Extremely strong assumptions on NTK/kernel regime results, however this seems to be fine even in the numerics.

---

> ### Author Rebuttal · Authors · 2026-03-31
>
> ## Figures and Tables: https://anonymous.4open.science/api/repo/attachment-7234/file/attachment.pdf
>
> > **Q1:** Experimental settings and eigenvalue trends are unclear.
>
> **A1:** In **Figure 2**, we use a two-layer neural network, vary the input dimension from $d=250$ to $500$, and fix the hidden width at $m=400$. The trends of the largest and smallest eigenvalues are included in **Figure 2A and Figure 3A** at the attachment `link`. We will include these settings more clearly and release the code to ensure reproducilibity after acceptance.
>
> > **Q2:** Do GLUs require more runtime because of more parameters?
>
> **A2:** Please see our response `A2` to reviewer `7bXQ`.
>
> > **Q3:** NTK assumptions are very strong, however this seems to be fine even in the numerics.
>
> **A3**: Please see our response `A2` to reviewer `Lk7y`.
>
> > **Q4:** How is "stochastic error" defined empirically?
>
> **A4:** Here, "stochastic error" refers to the gap between population loss and empirical loss, i.e., $L_D-L_S$. Empirically, we use test loss as a proxy for population loss, so the stochastic error is computed as test loss minus train loss. We will clarify this in the revision.
>
> > **Q5:** Where is the condition $d+1<n$ used?
>
> **A5:** Condition $d+1 < n$ means that the input sample dimension is less than the size of the dataset, which is a mild assumption in the over-parameterized regime. It ensures that the associated Wishart matrix $W = \frac{1}{d}XX^{\top}$ is singular with probability 1, see line (254). This simplifies our analysis in main text and keeps a clean form of the smallest eigenvalue of the NTK matrix.
>
> > **Q6:** Would Muon become less useful when GLU already improves geometry?
>
> **A6:** We compared SwiGLU and SiLU under both AdamW and Muon on three model families, see **Table 3A and Figure 4A** at the attachment `link`. From the result we can see that **when replacing AdamW with Muon, the optimization advantage of GLU becomes much smaller**. For example, the train loss of AdamW on Qwen changes from 2.18 to 2.07, while for Muon it changes from 1.83 to 1.86.
>
> Therefore, the answer is **yes**, if a substantial part of GLU's benefit comes from improving training geometry, then a more geometry-aware optimizer can reduce the marginal gain of GLU. We believe this is also a promising direction that worthy of further investigation.
>
> > **Q7:** Why are there no proofs for other activations? Is this strictly due to the availability of the arc-cosine kernel?
>
> **A7:** Yes, the main obstacle is the lack of **closed-form kernel expressions** analogous to the arc-cosine kernel for ReLU activation.
>
> However, we emphasize that the key structural insight regarding the NTK matrix, specifically the relation $\tilde K \approx K \odot (XX^{\top} / d)$, **is inherently independent of the specific choice of activation function**. This result arises from the architectural properties of GLUs rather than the precise functional form of the non-linearity.
>
> Furthermore, our empirical evaluations across various activation functions consistently support our theoretical claims. Specifically, experiments demonstrate that GLU-based models exhibit a **significantly lower condition number** compared to their non-GLU counterparts (as shown in **Figure 3** in the main text, **Table 2A** at the attachment `link`), and **superior convergence acceleration** across all tested settings (**Figure 6** in the main text, **Figure 1A and Figure 4A** at the attachment `link`).
>
> > **Q8:** How is Equation (16) derived from Theorem B.3?
>
> **A8:** First, we clarify a typo: Corollary B.3 was mistakenly written as Theorem B.3 in our text.
>
> Let $A = \tilde{\alpha}(XX^{\top})\odot (XX^{\top})$ and $B=\tilde{\beta}d XX^{\top}$. Then Equation (16) can be rewritten as
> $$
> \max\\{\lambda_1(A), \lambda_1(B)\\} \leq \lambda_1(A+B) \leq \lambda_1(A) + \lambda_1(B).
> $$
> The upper bound follows by setting $k=1$ in Corollary B.3, which gives $\lambda_1(A+B) - \lambda_1(A) \leq \lambda_1(B)$.
>
> For the lower bound, we use the fact that both $A$ and $B$ are positive semidefinite (see Line 875-876, which is right above Eq.(16)), so $\lambda_n(A) \geq 0$ and $\lambda_n(B) \geq 0$. Applying Corollary B.3 again with $k=1$ yields
> $$
> \lambda_1(A+B) \geq \lambda_1(A) + \lambda_n(B) \geq \lambda_1(A),
> $$
>
> $$
> \lambda_1(B+A) \geq \lambda_1(B) + \lambda_n(A) \geq \lambda_1(B).
> $$
> Hence $\lambda_1(A+B) \geq \max\\{\lambda_1(A), \lambda_1(B)\\}$.
>
> We will clarify this derivation in the revision.
>
> > **Q9:** Limitations should be discussed a bit more: e.g. NTK analysis done for 2 layer MLP only, when NTK assumptions break down, the infinite width case versus the actual implemented cases.
>
> **A9:** Thank you for your suggestion! We refer the reviewer to our response `A2` to reviewer `Lk7y` for discussions on the NTK assumptions. We will also include the discussion on these limitations more explicitly in the revision.

---

> > ### Author Rebuttal · Reviewer_kQfg · 2026-03-31
> >
> > Thanks for the detailed response and additional compute; I found the Muon comparison especially interesting.

---

> > > ### Author Response · Authors · 2026-04-01
> > >
> > > Thank you very much for your positive assessment of our analysis. We are glad that our response has addressed your concerns, and we will make these points clearer in the future revision.

---

### Official Review · Reviewer_Lk7y · 2026-03-13

**Soundness:** 2
**Presentation:** 2
**Significance:** 1
**Originality:** 1
**Overall Recommendation:** 3
**Confidence:** 4

**Summary:**

This paper provides an analysis of why GLU variants (ReGLU, SwiGLU etc.) consistently outperform standard FFN activations in practice. The paper studies two-layer networks in the NTK regime and derive an approximate Hadamard product relationship between the GLU and non-GLU NTK matrices. From this, the authors show that the GLU NTK has a smaller condition number, which implies faster convergence. They then analyze the resulting training dynamics, deriving a loss-crossing phenomenon where non-GLU models converge faster initially, but are eventually overtaken by GLU models. Finally, they present empirical evidence that GLU does not significantly reduce the so-called “stochastic error”, arguing the benefit is purely on the optimization side.

**Compliance With Llm Reviewing Policy:**

Affirmed.

**Final Justification:**

Although the original manuscript showed significant issues in presentation, unstated assumptions, and technical soundness/empirical work, the authors made a respectable effort during the rebuttal to address some of these original concerns. To me, the work still lacks significance and originality compared to earlier work by Liu et al. (2025), which is why I am leaning towards reject still.

**Key Questions For Authors:**

1. Regarding the stochastic error, I was a bit surprised to read that the error is not normalized. A relative measure like $(\mathcal L_D - \mathcal L_S) / \mathcal L_S$ would be more intuitive to me, as this removes the scale of the loss. Do you think that could have affected your results?

 2. Can the authors more explicitly articulate what technical challenges arose in the GLU setting that were not already addressed by Liu et al.'s framework?

3. Could the authors provide some (possibly qualitative) intuition behind the modelling assumptions made in §3, which were raised above?

**Limitations:**

yes

**Strengths And Weaknesses:**

Strengths:

- The paper contributes to the interesting domain of deriving theoretical explanations for why certain neural network components yield improved performance. The comparison between GLU structures and non-GLU structures is particularly interesting due to *GLUs’ lack of convincing motivation beyond empirics.

Weaknesses:

- To my reading, this seems like non-novel/very incremental work with respect to [1], which showed that ReLU activation leads to better feature separation (larger angle separation for similar data in the gradient feature space) and better NTK conditioning over linear using essentially the same tools/framework: Marchenko-Pastur, Weyl's inequality, arc-cosine kernel expansions, NTK condition number analysis, model gradient angle interpretations (Eq. 8).

- Some of the assumptions made during the presentation should be better justified to support claims about the applicability of the findings to ReGLUs in larger NNs. For instance, inputs to ReGLU are assumed to be Gaussian i.i.d. From $\mathcal N(0, I)$, which is generally not the case. Furthermore, the main theoretical results come from analyzing a two-layer network; the NTK of a deep network may have a qualitatively different structure from a two-layer one. The NTK framework requires $m \to \infty$, where the network is linearized around initialization and does not learn features. The networks where GLU matters in practice operate in the feature-learning regime, not the lazy/kernel regime.

- Although the paper's contributions are rather theoretical, some of the empirical work is not very convincing to me. The paper motivates itself by the dominance of SwiGLU in modern LLMs, but the only language model tested is GPT-2. Similarly, Fig. 3 shows condition numbers decrease with GLU on ViT and GPT-2, but this does not verify the paper's actual theoretical contribution that $\tilde{K} \approx K \odot (XX^\top/d)$. This could be an area of expansion that could greatly help the manuscript.

- Some presentation issues: the plot in Fig. 5 (and possibly Fig. 2) looks like a schematic figure, or with highly smoothed curves; if so, they should be labeled accordingly. Fig. 3 is a mixture between a barplot and a scatterplot, making readability difficult; I suggest simply using a boxplot. Some of the captions could be more informative (e.g., Fig. 4; what models?).

Minor/typos:

- In my opinion, the proof sketch outlined in §3.2 unnecessarily bloats the text and would benefit from being kept at a higher level, moving more of the derivation to the appendix.
- The term “stochastic error” is not standard and slightly misleading. I think a more common term is simply the “generalization gap”.
- Some typos: Corollary 4.2 2) “ReGLU model takes over and converges faster than ReGLU model” should be “ReLU model”, many missing articles (e.g., §3.1 title, L18 right column, …), L368: “tranjectory”


[1] Liu et al. (2025), NeurIPS: Better NTK Conditioning: A Free Lunch from (ReLU) Nonlinear Activation in Wide Neural Networks

---

> ### Author Rebuttal · Authors · 2026-03-31
>
> ## Figures and Tables: https://anonymous.4open.science/api/repo/attachment-7234/file/attachment.pdf
>
> > **Q1:** Non-novel work w.r.t Liu et al. (2025).
>
> **A1:** We agree that both works connect better NTK conditioning with faster optimization. However, **the core object and technical route are fundamentally different**.
>
> First, the **core object is different**. We study how **multiplicative gating** in GLU reshapes the NTK and affects the optimization speed and stochastic error. Our key object is the GLU-specific relation $\tilde K \approx K \odot (XX^\top/d)$. In contrast, their work mainly focus on the advantage of ReLU activation against linear network.
>
> Second, the **technical routes are different**. In Liu et al., only the first layer is trainable, and they use Rayleigh-quotient-based method to obtain comparative result. In our setting, all layers are trainable and the extra data-dependent factor $Px$ leads to an NTK of the form $\tilde K \approx K \odot (XX^\top/d)$. **This Hadamard reweighting breaks the monotonic spectral argument, so their proof technique cannot be directly transferred.** Instead, we use a different route based on **random matrix analysis** and kernel expansions.
>
> Third, we go beyond a conditioning comparison by analyzing the loss-crossing phenomenon, and by empirically showing that GLU's main advantage is on optimization rather than on the generalization gap.
>
> > **Q2:** Assumptions should be discussed, including Gaussian i.i.d. inputs, NTK structure beyond two-layer network and feature-learning regime.
>
> **A2:** We agree that the scope of the assumptions in Section 3 should be stated more clearly.
>
> First, for the **Gaussian i.i.d. input assumption**, it is used mainly for analytical tractability in the spectral analysis. The relation $\tilde K \approx K \odot (XX^\top/d)$ itself is derived for the two-layer GLU model **without requiring Gaussian inputs**. Importantly, once this relation holds, the factor $XX^\top/d$ suppresses off-diagonal correlations and makes $\tilde K$ more diagonally dominant than $K$. In the idealized case where the off-diagonal entries vanish, according to extreme value theory, for bounded Weibull distribution, the condition number is close to 1. Hence, **The role of the Gaussian assumption in our paper is therefore to make this intuition quantitative, rather than to make the mechanism itself valid.** We empirically validate this relation in **Table 1A** at the attachment `link`: even at width 2048, the Pearson correlation is already 0.994 on Gaussian data and 0.996 on MNIST, and the relative Frobenius error further decreases to 0.041 and 0.033 with larger width.
>
> Second, the same conditioning improvement persists well beyond Gaussian synthetic inputs and two-layer models, for example, the condition number drops from 7.89 to 1.82 on DeepSeekV3 with TinyStories, see **Table 2A** at the attachment `link`. This suggests that **the phenomenon is not a Gaussian-only artifact**.
>
> To further address the case beyond two-layer models, we also theoretically analyzed a model with preceding feature module $u=g(x)$, which is explained in detail in our response `A4` to reviewer `vHko`.
>
> Finally, regarding the **NTK regime**, our claim is not that NTK fully explains practical GLU training, but that it reveals the optimization-side mechanism: **GLU reshapes the kernel spectrum and improves conditioning**. Moreover, if we use mean field theory which operates in the feature learning regime, further analysis will also lead to a factor proportional to $x^{\top}x'/d$. Therefore, the Hadamard-type reweighting is not merely a lazy-regime artifact.
>
> > **Q3:** Insufficient experiments. The only language model tested is GPT-2, and the kernel relation is not tested.
>
> **A3:** We extended the evaluation to recent LLM families including Qwen, Llama, and DeepSeek. Across both random-token and TinyStories inputs, **GLU variants consistently yield lower condition numbers than their non-GLU counterparts** (For example, on random-token inputs, the condition number drops from 5.81 to 1.81 on Qwen3.5), **and also smaller training loss**, see **Table 2A and Figure 1A** at the `link`.
>
> We also directly validate the kernel relation $\tilde K\approx K\odot (XX^{\top}/d)$, see **Table 1A** at the `link`.
>
> > **Q4:** Stochastic error should be normalized.
>
> **A4:** We re-plotted all figures in Section 5 and Appendix D using the relative error $(L_D - L_S)/L_S$; see **Figure 4A** and **Figures 6A-10A** at the attachment `link`. The same conclusion still holds: GLU and non-GLU exhibit similar generalization gap.
>
> > **Q5:** Other issues: (presentation) Fig.5 - correct labeling, Fig.3 - use boxplot and improve captions; (minor) proofs should be kept at high level, use "generalization gap", fix typos.
>
> **A5:** We will revise these issues accordingly.

---

> > ### Author Rebuttal · Reviewer_Lk7y · 2026-04-03
> >
> > I highly appreciate the authors' efforts to add experiments to support evidence around the Gaussian i.i.d. assumption, adding more models to the evaluation, as well as providing results in terms of the normalized generalization gap. The results seem consistent. I am happy to increase my overall assessment.
> > My subjective assessment is still that this work is incremental to the Liu et al. 2025 paper, as described in the original review; I am not convinced that the approaches are "fundamentally different". Thus, I am still leaning towards rejection.

---

> > > ### Author Response · Authors · 2026-04-03
> > >
> > > We thank the reviewer for carefully reading our rebuttal, recognizing that the new experiments are consistent, and updating the score.
> > >
> > > That said, we would like to make the clarification: the statement that our work uses “essentially the same tools/framework” as Liu et al. (2025) is **a misunderstanding**. In fact,
> > >
> > > > **Their technical route cannot lead to our conclusion.**
> > >
> > > Due to character limit, we are unable to thoroughly clarify this in our initial response, so below we explain in further detail.
> > >
> > > ### 1. What did Liu et al. actually do?
> > >
> > > The aim of their work is to compare the spectral property of linear model with ReLU activated MLP.
> > >
> > > For **linear model**, using **matrix calculus**, their analysis gives an NTK matrix of the form $K = XX^{\top}$.
> > >
> > > For **ReLU model**, using **matrix calculus**, their analysis gives $K'\_{ij} = \sum_{k,l}A_{ikl}'A_{jkl}'$, with $A\_{ikl}':=\sqrt{2}X\_{ik}\mathbb{I}\_{li}$ and $\mathbb{I}_{li}:=\mathbb{I}\\{W_l^{\top}X_i\geq 0\\}$. Here $W$ is the parameter matrix of the first layer.
> > >
> > > Then, they **directly use the property of Rayleigh quotient**, which gives (Proof of **Lemma E.4** in Appendix G.5):
> > > $$
> > > \lambda_{\min}(K') = \min_{u\neq 0} \frac{u^{\top}K'u}{\\|u\\|^2}
> > > $$
> > >
> > > $$
> > > = \min_{u\neq 0} \frac{2\sum_{k,l}\sum_i(u_iX_{ik}\mathbb{I}_{li})^2}{\\|u\\|^2}
> > > $$
> > >
> > > $$
> > > = \min_{u\neq 0} \frac{2\sum_l(u_l')^{\top}K(u_l')}{\\|u\\|^2}
> > > $$
> > >
> > > $$
> > > \> \min_{u\neq 0} \frac{2\sum_l\\|u_l'\\|^2}{\\|u\\|^2}\cdot\lambda_{\min}(K)
> > > $$
> > >
> > > $$
> > > = \min\_{u\neq 0} \frac{\sum\_i u\_i^2\cdot 2\sum\_l \mathbb{I}\_{li}}{\sum_iu_i^2} \cdot \lambda_{\min} (K)
> > > $$
> > >
> > > $$
> > > = \lambda_{\min} (K).
> > > $$
> > >
> > > and vice versa for $\lambda_{\max}(K') < \lambda_{\max}(K)$, which lead to the conclusion that ReLU models are better conditioned than linear models.
> > >
> > > As is shown in this process, the main theoretical tools Liu et al. use are **(1) matrix calculus** to obtain the NTK matrix of the two models, and **(2) the property of Rayleigh quotient** to obtain a *comparative* result between the two eigenvalues. They
> > >
> > > > **do not use random matrix theory**, let alone random matrix tools including Marchenko-Pastur distribution, Weyl's inequality and Wishart matrix asymptotics that **the reviewer originally thought Liu et al. use**.
> > >
> > > Additionally, Liu et al. fix the second layer and only let the first layer to be trainable, and their method is not able to obtain the approximate expression for $\lambda_{\max}, \lambda_{\min}$.
> > >
> > > ### 2. Why does their route fail in our setting?
> > >
> > > Our key object is
> > > $$
> > > \tilde K \approx K \odot (XX^\top/d),
> > > $$
> > > **Once such a Hadamard factor appears, the problem can no longer be solved with Rayleigh quotient**.
> > >
> > > To see this, we have:
> > > $$
> > > \lambda\_{\min}\left(K\odot XX^{\top}/d\right) = \min_{u\neq 0} \frac{u^{\top}(K\odot XX^{\top})u}{d\\|u\\|^2} = \min_{u\neq 0} \frac{\sum_{i,j}u_iu_jK_{ij}(X_i^{\top}X_j)}{d\\|u\\|^2}.
> > > $$
> > > To continue this analysis, we must find a new vector $u'$ such that $u_i'u_j' = u_iu_j(X_i^{\top}X_j)$ for **any** dataset $X$, so that we can obtain the form $u'^{\top}Ku'$ in the numerator of Rayleigh quotient of $K$. However,
> > >
> > > > **This can only be achieved when $X_i \parallel X_j$, i.e., all samples of the dataset must be on the same line!**
> > >
> > > ### 3. What do we do instead?
> > >
> > > Therefore,
> > >
> > > > **Our route must be beyond the simple Rayleigh quotient analysis, hence we must adopt random matrix theory**.
> > >
> > > What we do is first derive the **GLU-specific kernel structure itself**. Starting from the NTK formulas of the ReLU and ReGLU two-layer models, after obtaining the kernel relation, we further rewrite
> > > $$
> > > K = \alpha XX^\top + \beta rr^\top + \gamma D,
> > > $$
> > > which gives the explicit decomposition of GLU model
> > > $$
> > > \tilde K =
> > > \frac{\alpha}{d}(XX^\top)\odot(XX^\top) +
> > > \frac{\beta}{d}(rr^\top)\odot(XX^\top) +
> > > \frac{\gamma}{d}D^2.
> > > $$
> > > This converts the problem into the spectral analysis of a **Hadamard-reweighted random matrix object**.
> > >
> > > **At this point, random matrix analysis comes in**. Under the Gaussian setting, $XX^\top/d$ is a **Wishart-type matrix**, whose eigenvalue distribution is characterised by **Marchenko-Pastur law**.
> > >
> > > For the *largest eigenvalue*, our **Proposition B.6** combines the decomposition above with explicit bounds for the rank-one update $rr^{\top}$ and Hadamard terms.
> > >
> > > For the *smallest eigenvalue*, our **Proposition B.9** uses the Schur product theorem together with our analysis on Hadamard powers of Wishart matrices $\frac{1}{d^2}(XX^{\top})\odot(XX^{\top})$, including **a recent random matrix result obtained by Pandit et al. (2024)**, to control the lower edge of the spectrum.
> > >
> > > In conclusion, Liu et al. only need a **Rayleigh quotient type** comparison argument because they only study whether linear model is better conditioned than ReLU model. We instead have to analyze a new Hadamard-reweighted kernel as **a random matrix object** due to the gating mechanism introduce by GLU, which is much more complicated than Liu et al.

---

### Decision · Program_Chairs · 2026-04-30

**Decision:**

Accept (regular)

**Comment:**

# Meta-review: 9594 — The Devil is in the Condition Numbers: Why is GLU Better than non-GLU Structure?

**Recommendation:** Weak Accept

The paper analyzes why GLU-family FFN blocks outperform their non-gated counterparts through an NTK-regime analysis of two-layer networks. The core technical result is an approximate Hadamard identity $\tilde K \approx K \odot (XX^\top/d)$ between the GLU and non-GLU NTKs, together with random-matrix bounds giving a factor-$d$ improvement in the NTK condition number. This is used to explain a loss-crossing phenomenon — non-GLU wins early, GLU wins asymptotically — and motivates the further empirical finding that the generalization gap is statistically indistinguishable between GLU and non-GLU models on ViT and GPT-2, suggesting GLU's benefit is optimization-side. Reviewer opinion was split, but after reading the rebuttal and the public discussion I lean toward acceptance.

The camera-ready should incorporate the following revisions, all tied to specific rebuttal commitments:

- **Rework Section 5 (stochastic error analysis).** This is the manuscript's weakest section, flagged as weak or speculative by three of the four reviewers, including the positive ones. Replace the unnormalized $L_D - L_S$ with the normalized gap $(L_D - L_S)/L_S$ (authors already re-ran these plots during rebuttal and the conclusion survived), adopt the standard "generalization gap" terminology, and reframe the section as a focused ablation rather than as a primary contribution.
- **State the parameter-count control explicitly.** The submitted manuscript does not describe how GLU and non-GLU runs are matched on parameter count; the authors clarified in rebuttal that they reduce the intermediate dimension in GLU models to keep total parameters comparable (following Shazeer 2020 and Narang et al. 2021). This must be stated explicitly in the main text alongside the training-loss comparisons.
- **Fold in the deeper-model kernel extension.** The rebuttal derivation $\tilde K \approx K \odot (UU^\top/d) + K_g \odot (\Sigma_\phi/d)$ for models with a preceding feature module $u = g(x)$, empirically validated for both MLP and attention-based feature modules, materially strengthens the applicability of the main result and should be folded into the paper (main text or a clearly marked appendix), along with the α-interpolation experiment that continuously deforms non-GLU into GLU and shows monotone decrease in both training loss and condition number.
- **Elevate the direct Hadamard kernel-relation validation.** The rebuttal verified $\tilde K \approx K \odot (XX^\top/d)$ directly (Pearson $>0.99$ and relative Frobenius error $\sim 0.03$ at width 2048 on both Gaussian and MNIST inputs). This belongs in the main body alongside Figure 3, which currently only shows condition-number drops, not the kernel identity that drives the theory.
- **Minor revisions.** Fix the "ReGLU" → "ReLU" typo in Corollary 4.2, label Figures 2 and 5 as schematic vs. empirical where applicable, and tighten the Figure 4 caption.